# In-Context Fine-Tuning for Time-Series Foundation Models

**Matthew Faw** [1]   **Rajat Sen** [2]   **Yichen Zhou** [2]   **Abhimanyu Das** [2]

## Abstract

Motivated by the recent success of time-series foundation models for zero-shot forecasting, we present a methodology for *in-context fine-tuning* of a time-series foundation model. In particular, we design a pretrained foundation model that can be prompted (at inference time) with multiple time-series examples, in order to forecast a target time-series into the future. Our foundation model is specifically trained to utilize examples from multiple related time-series in its context window (in addition to the history of the target time-series) to help it adapt to the specific distribution of the target domain at inference time. We show that such a foundation model that uses in-context examples at inference time can obtain much better performance on popular forecasting benchmarks compared to supervised deep learning methods, statistical models, and other time series foundation models. Interestingly, our in-context fine-tuning approach even matches the performance of a foundation model that is explicitly fine-tuned on the target domain.

## 1. Introduction

Time-series data is ubiquitous in several domains such as retail, finance, manufacturing, healthcare, and the natural sciences. In many of these domains, time-series forecasting, i.e., predicting time-series into the future, is a critical problem – for example, in applications like retail forecasting, climate and weather predictions, and traffic forecasting. In the last decade, deep learning approaches (Salinas et al., 2020; Oreshkin et al., 2020; Sen et al., 2019) have become popular in forecasting, often outperforming statistical approaches like ARIMA (Box & Jenkins, 1968). However, until recently, deep learning approaches for forecasting have ad-

hered to the traditional supervised machine learning framework of having to train a forecasting model on task-specific training data, before being able to perform forecasting for that task. On the other hand, in Natural Language Processing (NLP), Large Language Models (LLMs) (Radford et al., 2019; Brown et al., 2020) have shown the promise of foundation models: a single pretrained model can perform well and adapt to tasks like translation, code generation during inference time in a zero-shot or few-shot manner.

Motivated by the success in NLP, there has been significant work in recent years on time-series foundation models for forecasting, ranging from re-purposing LLMs directly for forecasting (Gruver et al., 2023) to fine-tuning pretrained LLMs on time-series data (Zhou et al., 2023; Chang et al., 2023) to pretraining time-series foundation models from scratch (Das et al., 2024; Goswami et al., 2024; Woo et al., 2024; Ansari et al., 2024; Garza & Mergenthaler-Canseco, 2023). The last approach, in particular, has been shown to obtain strong zero-shot accuracy, rivaling the best supervised models trained specifically for the target datasets.

Several of these papers (Zhou et al., 2023; Ansari et al., 2024; Goswami et al., 2024) have shown an opportunity for further accuracy improvement by fine-tuning of their pretrained models on target datasets. However, this breaks the zero-shot paradigm that precisely makes these time-series foundation models so appealing to practitioners who do not want to build training pipelines. This raises a natural question: *Can we recover the benefits of fine-tuning a time-series foundation model by providing examples from a target dataset at inference time?*

At the same time, the first generation of time-series foundation models lack some of the desirable features of LLMs with respect to *in-context learning*: the zero-shot performance of an LLM can be greatly improved *at inference time* by using its context window for prompting techniques such as few-shot (Brown et al., 2020), chain-of-thought (Wei et al., 2022b) or instruction tuning (Wei et al., 2022a). These papers have shown emergent in-context learning abilities for LLMs. In particular, if we prompt them with related examples and demonstrations then ask a specialized question, the model is able to reason similarly for the question at hand.

In this work, we study a methodology to enable similar in-context ability for time-series foundation models i.e. being

---

[1]Georgia Institute of Technology. Part of this work was done while the author was a Student Researcher and Visiting Researcher at Google Research. [2]Google Research. Correspondence to: Matthew Faw <mfaw3@gatech.edu>.

able to prompt the model with time-series examples from the target domain, and recover the benefits of domain-specific fine-tuning. We refer to this as *in-context fine-tuning*.[1]

We train a foundation model that lets us forecast a time-series by providing in its context window not just the historical values of the time-series, but also examples from other related time-series that could help the model adapt, *at inference time*, to the distribution of the target time-series. For example, consider a highway traffic prediction system that stores hourly data from the last week, in order to predict the future hourly traffic for a particular highway. Consider a time-series foundation model that has not seen data in pretraining that captures the temporal patterns in this traffic data. Then, simply prompting the model with the previous week's traffic time-series for that highway might not be enough to obtain accurate zero-shot performance. However, adding to the prompt historical traffic data from other highways and weeks, might help the model better adapt to the traffic data distribution.

The main contributions of our paper are as follows:

**(i)** We introduce the study of in-context fine-tuning for time-series foundation models, and propose the use of prompts that not only include the usual history of the target time-series for forecasting, but also include related time-series examples in-context.

We propose a methodology for training such a model by starting from a base time-series foundation model and continue pretraining it with in-context examples. Our training is decoder-only (Liu et al., 2018) and can adapt to varying history and horizon lengths (up to a certain maximum history) and to a varying number of related time-series examples in the context window (again up to a certain maximum number of examples). The resulting model can then learn to borrow patterns from these related examples to perform better on the target forecasting task.

**(ii)** We empirically evaluate the benefits of in-context fine-tuning using our foundation model, and show that in-context fine-tuning can lead to better zero-shot performance on popular forecasting benchmarks as compared to supervised deep learning methods, statistical models as well as other foundation models. In particular, on a well known forecasting benchmark, comprised of 23 datasets not included in the pretaining of our foundation models, we show that our in-

---

[1]Terminology: In the LLM domain, this notion is also called "few-shot learning" (Brown et al., 2020), "few-shot prompting" (Ye & Durrett, 2022), or "in-context tuning" (Chen et al., 2022). Also, borrowing from LLM literature, we will refer to the generic ability of pretrained foundation models to learn from information in their context window at inference time as "in-context learning". Additionally, we will refer to pretrained models that do not need gradient-updates via explicit training or tuning for an unseen target dataset as "zero-shot".

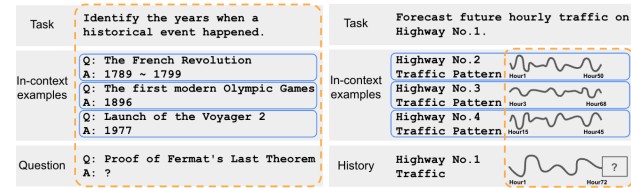

*Figure 1.* Analogous to few-shot prompting of a foundation LLM (left), we train a time-series foundation model to support few-shot prompting with an arbitrary number of related in-context time-series examples (right). The dashed box encloses the full context window/prompt.

context fine-tuned model is 6.8% better than the base model we start from, while also being 5% better than the next best baseline. More importantly, (and perhaps surprisingly), it *achieves similar performance to that obtained by explicitly fine-tuning the base model* on the training split of every dataset in the benchmark.

## 2. Related Work

As mentioned previously, there has been a spurt of recent work on time-series foundation models for forecasting. These approaches can be broadly divided into three categories. (i) Prompting LLMs like GPT-4 to directly predict the future of a numerical series encoded as text. This was investigated in LLMTime (Gruver et al., 2023); despite the initial promise subsequent works have shown that such approaches can be lacking in accuracy (Woo et al., 2024; Das et al., 2024). (ii) Fine-tuning pretrained LLMs like GPT2 on time-series data with adapter layers (Zhou et al., 2023; Chang et al., 2023). These approaches have mostly been shown to work well in the dataset-to-dataset transfer learning setting (rather than in the zero-shot setting), and they are also disadvantaged from having to use excessively large models due to their LLM backbones. (iii) Pretraining transformer based models from scratch on huge volumes of time-series data, which seems to be the most promising approach towards times-series foundation models (Das et al., 2024; Garza & Mergenthaler-Canseco, 2023; Ansari et al., 2024; Woo et al., 2024; Goswami et al., 2024). Indeed, some of these models have shown superior zero-shot accuracy when compared to supervised deep forecasters even on datasets that are outside of their pretraining set.

Some of the above papers, e.g., (Ansari et al., 2024; Goswami et al., 2024), have additionally shown that their pretrained models' performance can be further improved by fine-tuning the model on examples from the target dataset. While this supervised fine-tuning results in improved per-task accuracy, this practice breaks the zero-shot paradigm in terms of requiring extra training on the target dataset.

In the NLP domain, a defining property of a foundation

LLM is its ability to be further adapted to domain-specific tasks through either fine-tuning or prompting. In particular, LLMs have been shown to perform *in-context learning* on a variety of downstream NLP tasks by prompting them with a natural language instruction (Radford et al., 2019) and a few demonstrations or examples of the task. This phenomenon is also referred to as *few-shot learning* (Brown et al., 2020). Subsequent works (Min et al., 2022a; Chen et al., 2022) have proposed fine-tuning a pretrained LLM to obtain better performance on few-shot learning prompts. Other papers (Min et al., 2022b; Wei et al., 2023) have empirically investigated how few-shot learning works in LLMs. More recently, Shi et al. (2024) explored a similar idea for in-context pretraining, where they pretrain an LLM on sequences of related documents. This in-context learning ability is widely recognized as being associated with the stacked transformers used in the LLMs, and their theoretical properties are studied in a more precise sense (Garg et al., 2022; Von Oswald et al., 2023; Ahn et al., 2024) for simpler function classes such as linear regression.

Despite the commonality between time-series foundation models and LLMs, it is not obvious how (or even if) the phenomenon of few-shot learning for NLP tasks carry over to the time-series setting. There is no clear definition of few-shot learning in terms of a time-series foundation model. In fact, prior pretrained time-series foundation models like (Ansari et al., 2024; Das et al., 2024; Garza & Mergenthaler-Canseco, 2023) do not provide a clear way for prompting with anything other than the past values of a time-series in the context window. The MOIRAI model (Woo et al., 2024) supports the functionality of any-variate forecasting, which allows the model to take as input arbitrary number of variates (up to a certain maximum). While one could provide in-context examples as additional variates, their approach is mostly aimed at multi-variate datasets. Nevertheless, we also compare with this model in Table 1.

## 3. Problem Definition

Time-series foundation models aim to build a general purpose forecaster that can take in a past *history* of a target forecasting task, $\mathbf{y}_{1:L} = \{y_1, y_2, \cdots y_L\}$, where we look back $L$ time-steps and map them to a forecast $\widehat{\mathbf{y}}_{L+1:L+H}$, for a horizon length of $H$. The aim is to have $\widehat{\mathbf{y}}_{L+1:L+H}$ as close as possible to the unseen future $\mathbf{y}_{L+1:L+H}$ according to some well defined error metric. Such a model can be thought of as a function,

$$g : \mathbf{y}_{1:L} \to \widehat{\mathbf{y}}_{L+1:L+H} \qquad (1)$$

which is capable for handling different values of $L$ and $H$.

In this work, we aim to further enhance the abilities of such models by enriching their context. In addition to the target task's history $\mathbf{y}_{1:L}$, we add up to $n-1$ *in-context*

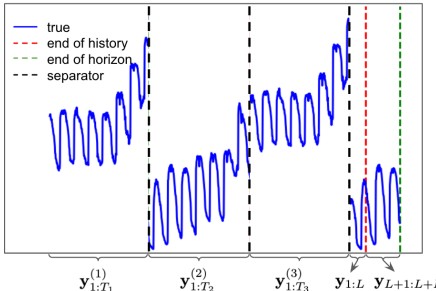

Figure 2. An example prediction task. The three black dashed lines (separators) separate the three in-context examples $\{\mathbf{y}_{1:T_i}^{(i)}\}_{i \in [3]}$ and the history $\mathbf{y}_{1:L}$. The goal is to predict the horizon $\mathbf{y}_{L+1:L+H}$ of the history $\mathbf{y}_{1:L}$.

*examples* of the form $\{\mathbf{y}_{1:T_1}^{(1)}, \mathbf{y}_{1:T_2}^{(2)}, \cdots \mathbf{y}_{1:T_{n-1}}^{(n-1)}\}$ that can represent the past time-points of other related time-series (with possibly varying lengths $T_1, \cdots, T_{n-1}$). In the case of our motivating example of highway traffic prediction, $\mathbf{y}_{1:L}$ is a week of hourly traffic data on that highway, and $\{\mathbf{y}_{1:T_1}^{(1)}, \mathbf{y}_{1:T_2}^{(2)}, \cdots \mathbf{y}_{1:T_{n-1}}^{(n-1)}\}$ are traffic data on $n-1$ nearby highways. We plot an example prediction task with three in-context examples in Figure 2.

Therefore, the enhanced forecasting problem is aimed at training a model $f$,

$$f : \left( \mathbf{y}_{1:T_1}^{(1)}, \mathbf{y}_{1:T_2}^{(2)}, \cdots \mathbf{y}_{1:T_{n-1}}^{(n-1)}, \mathbf{y}_{1:L} \right) \to \widehat{\mathbf{y}}_{L+1:L+H}. \qquad (2)$$

As before, our time-series foundation model should be able to handle different values of $L$ and $H$. Additionally it should be able to support any number of in-context examples $(n-1)$ ranging from zero to a maximum value. With some abuse of notation, let us index the target task's forecasting history and horizon as the $n$-th example i.e. $\mathbf{y}_{1:T_n}^{(n)} := \mathbf{y}_{1:L+H}$, where $T_n = L + H$. Therefore, our decoder-only model will work with $n$ examples of the form $\{\mathbf{y}_{1:T_1}^{(1)}, \mathbf{y}_{1:T_2}^{(2)}, \cdots, \mathbf{y}_{1:T_n}^{(n)}\}$ which are drawn from related time-series. Henceforth, we will refer to $\{\mathbf{y}_{1:T_i}^{(i)}\}_{i=1}^{n}$ as the *context* (synonymous with prompt) supplied to the model.

## 4. Model Architecture

Motivated by the strong zero-shot performance achieved by stacked transformer models in decoder-only mode for time-series forecasting, we propose to adapt a base TimesFM model (Das et al., 2024) to leverage the additional information available via in-context examples. In particular, we first pretrain TimesFM in its original fashion to obtain a base checkpoint *TimesFM (base)*. We then modify the model architecture and continue pretraining from TimesFM (base) using training data with in-context examples (we call

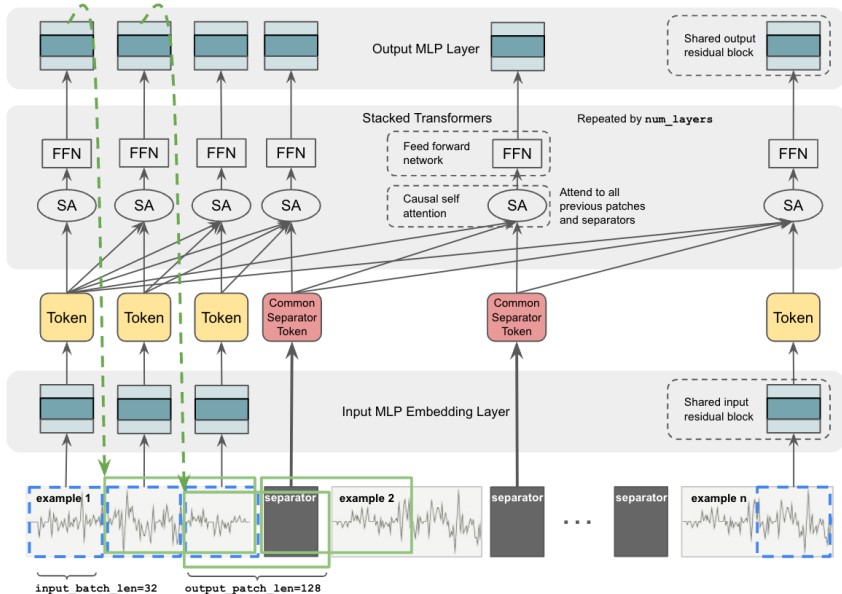

*Figure 3.* TimesFM-ICF employs the decoder-only architecture for time-series prediction with in-context examples.

this phase *continued pretraining*) to obtain a new pretrained foundation model *TimesFM-ICF*.

Adapting their model architecture to make use of the in-context examples is somewhat delicate, and requires modifications to the original model. A depiction of our proposed model architecture is given in Figure 3. As in their model, our model partitions each example into non-overlapping input *patches*, and uses a shared input residual block (a one-hidden layer perceptron with skip connection, see Das et al. (2023)), to embed each patch as a token before feeding the tokens into the stacked transformers in a decoder-only fashion. The output embeddings are mapped to the next output patches via another shared output residual block.

To teach the model to use the new in-context examples, we adapt the original TimesFM architecture to better handle (1) the in-context example separators, (2) the cross-example attention, and (3) the positional encoding (by applying no positional encoding when pretraining TimesFM (base), see Appendix A.3). Despite these changes, we are still able to leverage the TimesFM (base) checkpoint, which was pretrained for forecasting given just the history of the target time-series. We describe the key details of our architecture design below.

### 4.1. Separators for In-Context Examples

Our context window contains in-context examples from different time-series. Hence the model needs to be able to separate these, since naïve concatenation can confuse the model. Consider the example in Figure 4. If we naïvely concatenate multiple in-context examples (e.g., linear trends,

Figure 4a) together, then the combination of these trends may appear to the model as an entirely different time-series (e.g., a triangle wave, Figure 4b). Therefore, we choose to insert a common learnable separator token after each in-context example. We visually depict these separators as the dashed lines in Figure 4a. When feeding examples to the decoder, we sequentially pass each tokenized patch of each time-series example to the model, followed by the separator token at the end of an example; depicted in Figure 3.

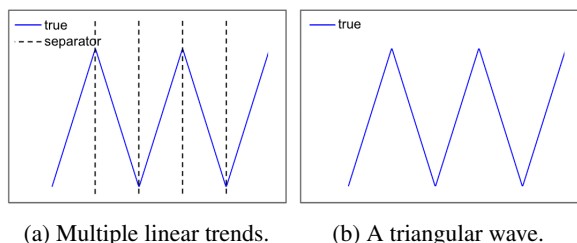

(a) Multiple linear trends.          (b) A triangular wave.

*Figure 4.* Concatenating in-context examples together without separators can confuse the model: multiple linear trends look like a triangular wave if concatenated naïvely.

### 4.2. Cross-Example Attention

In order to allow our model to distinguish between different in-context examples, we allow the transformer to attend (causally) to all previous patches including the separator tokens. Note that, if the model did not attend to the separator tokens, then we could never hope to distinguish between the two scenarios from Figure 4a and Figure 4b. By attending to the previous separator tokens, the model can potentially distinguish how many in-context examples have been pro-

cessed so far.

Although at the input to the stacked transformer we use a common separator token to separate the examples, the output tokens corresponding to the positions of these separator tokens can play a much more nuanced role as we proceed through the subsequent transformer layers. As the output tokens corresponding to these separator tokens causally attend to all previous tokens, after several transformer layers these tokens can, for instance, potentially summarize the information in all the patches corresponding to their example and/or convey the separation boundaries of the different in-context examples to the model.

### 4.3. Overall Model

Since our model builds upon the TimesFM architecture (Das et al., 2024), we introduce a similar notation style for ease of exposition. The model processes in-context examples in the following fashion. Starting with an example input $\{\mathbf{y}_{1:T_1}^{(1)}, \ldots, \mathbf{y}_{1:T_n}^{(n)}\}$, each example $\mathbf{y}_{1:T_i}^{(i)}$ is partitioned into input patches of length $p$:

$$\tilde{\mathbf{y}}_j^{(i)} = \mathbf{y}_{p(j-1)+1:pj}^{(i)} \quad \forall j \in [\lceil T_i/p \rceil] \text{ and } i \in [n].$$

As in (Das et al., 2024), our model takes an additional padding mask $\mathbf{m}_{1:T_i}^{(i)}$ to ensure that it makes good predictions on time-series which are not a multiple of the patch length $p$. Analogously to the partitioning of the example inputs, we partition the padding masks as:

$$\tilde{\mathbf{m}}_j^{(i)} = \mathbf{m}_{p(j-1)+1:pj}^{(i)} \quad \forall j \in [\lceil T_i/p \rceil] \text{ and } i \in [n].$$

Given these patches and masks, we feed each patch $\tilde{\mathbf{y}}_j^{(i)}$ through a common MLP embedding layer to obtain tokens:

$$\mathbf{t}_j^{(i)} = \mathsf{InputResidualLayer}(\tilde{\mathbf{y}}_j^{(i)} \odot (1 - \tilde{\mathbf{m}}_j^{(i)})).$$

We will slightly abuse notation by denoting the separator token $\boldsymbol{\sigma}$ as $\mathbf{t}_{\lceil T_i/p \rceil + 1}^{(i)} = \boldsymbol{\sigma}$, and let the mask for the separator token $\tilde{\mathbf{m}}_{\lceil T_i/p \rceil + 1}^{(i)} = \mathbf{0}$ (i.e., the separator tokens are never masked). Write $J_i = \lceil T_i/p \rceil + 1$, the total number of patches belonging to the example $\mathbf{y}_{1:T_i}^{(i)}$.

After tokenizing the input patches, we feed the tokens, together with a learnable separator token $\boldsymbol{\sigma}$, autoregressively to the stacked transformer layers in decoder-only mode. We take $\dot{m}_j^{(i)}$ to be the last entry of $\tilde{\mathbf{m}}_j^{(i)2}$, and denote the sequence of token/mask pairs corresponding to example $i$ as

$$\tilde{\mathbf{t}}_{1:j}^{(i)} = ((\mathbf{t}_1^{(i)}, \dot{m}_1^{(i)}), \ldots, (\mathbf{t}_j^{(i)}, \dot{m}_j^{(i)})) \quad \forall j \in [J_i].$$

---

[2]Intuitively, $\dot{m}_j^{(i)}$ indicates whether or not patch $\tilde{\mathbf{y}}_j^{(i)}$ is masked from the right. We attend only to patches which are not padded from the right, and have at least one unpadded values (see Appendix A.2)

Then, the output of the stacked transformer layer for token $\mathbf{t}_j^{(i)}$ can be described as:

$$\mathbf{o}_j^{(i)} = \mathsf{StackedTransformer}(\tilde{\mathbf{t}}_{1:J_1}^{(1)}, \ldots, \tilde{\mathbf{t}}_{1:J_{i-1}}^{(i-1)}, \tilde{\mathbf{t}}_{1:j}^{(i)}).$$

We emphasize the output $\mathbf{o}_j^{(i)}$ for token $\mathbf{t}_j^{(i)}$ defined above depends on (i) all previous (unmasked) tokens $\mathbf{t}_{j'}^{(i')}$, $i' < i$ and $j' \in [\lceil T_{i'}/p \rceil]$, (ii) the $i-1$ separator tokens $\mathbf{t}_{\lceil T_{i'}/p \rceil + 1}^{(i')} = \boldsymbol{\sigma}$ for $i' < i$, and (iii) the tokens $\tilde{\mathbf{t}}_{1:j}^{(i)}$ for the current example.

Finally, we feed the outputs $\mathbf{o}_j^{(i)}$ for each example $i$ with patch $j$ from the stacked transformer through a residual block to obtain the predicted time-series:

$$\hat{\mathbf{y}}_{pj+1:pj+h}^{(i)} = \mathsf{OutputResidualLayer}(\mathbf{o}_j^{(i)}).$$

This corresponds to the model's prediction of the next $h$ steps (output patch length) of $\mathbf{y}_{pj+1:pj+h}^{(i)}$.

We use the same loss function as original TimesFM model.

## 5. Continued Pretraining Data

As mentioned before, we start with TimesFM (base) which was pretrained on a diverse corpus of about 400B timepoints (see Table 3 in Appendix A.2 and Das et al. (2024) for more details on the datasets). We then continue pretraining it on training data containing in-context examples.

### 5.1. Context Generation

We convert individual datasets to generate *contexts* with in-context examples that the model sees during the continued pretraining. Recall that the original TimesFM model is trained up to a maximum history length of $L_{max} = 512$. During the training of TimesFM (base) a time-series of length $T = L_{max} + h$ is loaded for back propagation where $h = 128$ is the output patch length. Therefore, we choose $T$ as the maximum length of our $n$ in-context examples. For any time-series in a particular dataset, we use windowing with a shift of 1 to generate examples of length $T$ i.e. for a time-series $\mathbf{y}_{1:M}$ the possibles examples are $\{\mathbf{y}_{1:T}, \mathbf{y}_{2:T+1}, \cdots \mathbf{y}_{M-T+1:M}\}$. For time-series that are less than $T$ in length, we generate padded examples as detailed in Appendix A.2. Now these examples are packed in groups of $n$ to form the context. We consider two kinds of grouping:

*Times-series level:* For a long time-series, we can split the original time-series into shorter time-series examples, each of length $T$, then select $n$ of those shorter examples to form the context $\{\mathbf{y}_{1:T}^{(i)}\}_{i=1}^n$ for the original time-series.

*Dataset level:* For each dataset, we can group any $n$ segments of length $T$ from any of the time-series in that dataset,

to form a context. For instance, a set of $n$ segments from any of the time-series from the Electricity dataset could be grouped to form a context $\{\mathbf{y}_{1:T}^{(i)}\}_{i=1}^{n}$.

Both time-series level and dataset level groupings guarantee that the grouped examples have similar patterns to borrow from each other.

## 5.2. Dataset Mixture

We choose all datasets in Table 3 (the dataset list used to train TimesFM (base)) other than the four Wiki datasets to generate in-context examples for continued training. The Wiki datasets contain millions of time-series that correspond to a wide variety of articles, which need not be related to each other.

For the remaining datasets, we set the number of examples in each context as $n = 50$ and generate contexts from both time-series level and dataset level grouping. Note that if all the time-series in a dataset have a total of $N$ examples, then generating all $\binom{N}{n}$ such contexts is intractable. Therefore, we randomly generate $20N$ such groups of $n$ examples as our training contexts.

Following the original TimesFM paper, the training data loader samples 90% real data and 10% synthetic, with the real data mixture providing equal weights to the groups: hourly + sub-hourly, daily, weekly, and monthly datasets. Moreover, we provide equal weights to the two kinds of examples i.e., time-series level and dataset level.

## 5.3. In-Context Example Selection

For choosing the 50 examples to be added to the context, we adopt a very simple strategy of using 5 examples from the immediate history of the time-series and the remaining examples chosen at random from the history of other time-series in the same datasets. In Section A.9, we experiment with some other simple example selection strategies. While we leave a more detailed investigation to future work, our results show that even naive approaches like random selection and selecting examples from the immediate history are sufficient to obtain accuracy gains with in-context fine-tuning.

## 6. Experimental Results

Similar to prior works, we report our results on the Chronos zero-shot benchmarks from Ansari et al. (2024), as well as rolling-window evaluation of the ETT datasets (Zhou et al., 2021). No data from these datasets (not even the training splits) was used in the training of our base model TimesFM (base), or our in-context fine-tuned model TimesFM-ICF. Since TimesFM-ICF uses examples of length 512, we report the numbers of TimesFM (base) with a maximum history

length of 512 unless otherwise specified.

## 6.1. The Fine-Tuning-per-Dataset Baseline

In all our experiments, we also compare with an extremely strong baseline TimesFM-FT which is the TimesFM (base) model fine-tuned on the training split of each dataset and then evaluated on the corresponding test split. Our main goal is to study whether TimesFM-ICF can match or surpass the performance of this baseline but in a zero-shot manner (without any gradient updates during inference). In both the benchmarks in Sections 6.2 and 6.3, we perform two kinds of fine-tuning (i) Full: update all the weights of the models (ii) Linear Probe (LP): update just the input and output MLP layers. Then we report the numbers from the best of the two. The details of fine-tuning are provided in Appendix A.4.

## 6.2. Out-of-Domain Forecasting

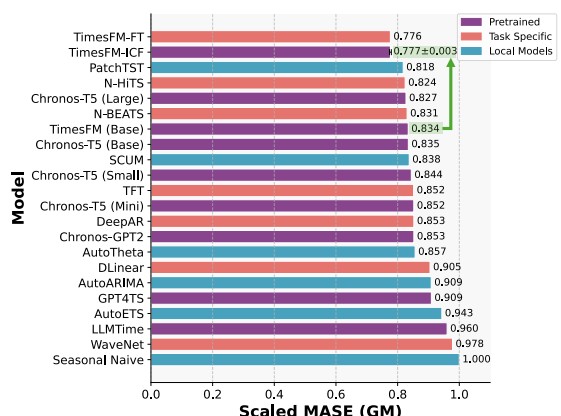

*Figure 5.* Geometric mean of scaled MASE on the OOD Benchmark. This benchmark is essentially the zero-shot benchmark used in (Ansari et al., 2024), modified slightly to guarantee a zero-shot evaluation of TimesFM-ICF. Our in-context fine-tuning approach improves the performance TimesFM (base) over all other benchmark models, and achieves the same performance as that of TimesFM-FT , the model which separately fine-tunes TimesFM (base) on the training split of each task before making predictions.

The Chronos zero-shot benchmark is a collection of 27 datasets of different training and prediction lengths that covers granularities ranging from minutes to years and domains including finance, demand forecasting, weather, and traffic. We report our results on the 23 datasets from this benchmark that were not used during training of TimesFM-ICF (or the base model). We will refer to this benchmark as the **OOD Benchmark** and provide more details in Appendix A.2.

The datasets contain time-series with vastly different scales, so we cannot aggregate the raw metrics. Therefore, following Ansari et al. (2024), we calculate the MASE for all methods and normalize them by the MASE achieved by a seasonal naive baseline that just repeats the last period's

values in the history for the whole horizon. Then we report the Geometric Mean of these scaled MASE values across all datasets. Note that when dealing with normalized metrics it is better to report the Geometric Mean (Fleming & Wallace, 1986). We borrow the official numbers for baselines from (Ansari et al., 2024) whenever possible. More details about the benchmark, error metrics and baselines are provided in Appendix A.6.

We include around 20 baselines including task specific (trained on the training set of each dataset and then evaluated) deep learning models such as DeepAR (Salinas et al., 2020), N-BEATS (Oreshkin et al., 2020) and WaveNet (van den Oord et al., 2016), statistical local (per time-series) models like as Exponential Smoothing(ETS) and ARIMA and other leading pretrained foundation models like Chronos-T5 (Ansari et al., 2024), and LLMTime (Gruver et al., 2023), more in Appendix A.2. Note that we are unable to compare with MOIRAI (Woo et al., 2024) because its pretraining data has more than 80% overlap with this benchmark. We extensively compare with MOIRAI on long horizon tasks in Section 6.3.

TimesFM-ICF can handle a maximum of 50 examples in its context. Whenever the time-series is long enough we include 5 within series examples (including the history of the time-series to be forecasted on) within the context and the rest of the examples are gathered randomly from other time-series in the same dataset (we ablate on a few other ways of choosing examples in Section A.9). Since there is randomness during inference, we average over 10 runs and report the standard error bar.

**Results.** We summarize our results in Figure 5. We can observe that unsurprisingly the per dataset fine-tuned TimesFM-FT is the strongest performer as it starts from a strong zero-shot model and is further finetuned to adapt to the dataset at hand. However, it can be seen that TimesFM-ICF can match the performance of TimesFM-FT completely out of the box at inference time, when supplied with in-context examples. Due to this ability TimesFM-ICF improves over TimesFM (base) by 6.8%. Moreover, TimesFM-ICF is better than the next best baseline, PatchTST by 5%.

Timing-wise, TimesFM-ICF directly utilizes in-context examples and TimesFM-FT needs to be fine-tuned per dataset. Although TimesFM-ICF requires more time to perform each forecast, it overall completes the OOD Benchmark 16x faster than TimesFM-FT does (25 minutes vs 418 minutes, see Appendix A.7).

### 6.3. Long Horizon Forecasting on ETT

A group of long horizon datasets have been commonly used for benchmarking (mainly) transformer based deep learning algorithms starting from (Zhou et al., 2021). Some of the datasets in these benchmarks are in our pretraining datasets (like Electricity and Traffic). Therefore, for the interest of zero-shot evaluation we use the 4 Electricity Transformer Temperature (ETT) datasets, specifically ETTh1, ETTh2 (hourly) and ETTm1, ETTm2 (15 min).

We conduct the same evaluation as in the long sequence forecasting evaluation (Woo et al., 2024) on these datasets, focusing on the task of predicting horizon lengths 96, 192, 336, and 720. We provide rolling validation numbers for the test time-period which consists the last 1/5-th of the time-points. This is standard for these benchmarks (Nie et al., 2023), where the datasets are split into train:validation:test in the ratio 7:1:2.

In addition to the evaluations on these datasets from (Woo et al., 2024, Table 6), we evaluate our TimesFM-ICF against TimesFM (base), and the TimesFM-FT model discussed in Section 6.1.

We present the MAE loss for each dataset, averaged over the four horizon lengths 96, 192, 336, 720, in Table 1. For a detailed breakdown of the MAE losses, see Table 9. Note that since the MAE is computed on scaled datasets in this benchmark (Zhou et al., 2021), we can directly report the arithmetic mean across datasets. We see that TimesFM-ICF rivals or outperforms TimesFM-FT which was finetuned explicitly on the target dataset's distribution. Moreover, TimesFM-ICF outperforms or equals the performance of all other baselines.

### 6.4. Ablation

#### 6.4.1. NUMBER OF IN-CONTEXT EXAMPLES

An important trade-off between speed and accuracy can be achieved by the number of in-context examples. We perform an ablation study of the same using the short context datasets in the OOD Benchmark i.e., datasets where we can only get one example per time-series (the one whose future we are predicting) and the rest of the examples are generated randomly from all the other time-series across the dataset. We perform the ablation over these datasets for two reasons (i) this removes the complication of the example selection strategy (studied separately later) (ii) short context datasets are where we know for sure that the positive effects in-context fine-tuning are coming from few-shot examples and not from just having a longer context (again studied separately in the next section). The short context datasets are listed in Table 5.

We plot the Scaled MASE (GM) vs number of in-context examples in Figure 6. The same figure also shows the total inference time[3]. The experiments are repeated 5 times and

---

[3]The inference numbers are reported on TPUv5e with 8 tensor cores.

*Table 1.* MAE of TimesFM-ICF against other supervised and zero-shot methods on ETT Rolling Window, averaged over forecast horizons $\{96, 192, 336, 720\}$. See Table 9 for a detailed breakdown. We bold the numbers which are the best in every row, and including the ones that are within standard error of the best.

| | Few-shot | Zero-shot | | | | Task-specific | | | | | | | |
| --- | --- | --- | --- | --- | --- | --- | --- | --- | --- | --- | --- | --- | --- |
| Dataset | TimesFM-ICF | TimesFM (Base) | Moirai (Small) | Moirai (Base) | Moirai (Large) | TimesFM-FT | iTransformer | TimesNet | PatchTST | Crossformer | DLinear | SCINet | FEDformer |
| ETTh1 | **0.405** | 0.417 | 0.424 | 0.438 | 0.469 | 0.407 | 0.447 | 0.450 | 0.454 | 0.522 | 0.452 | 0.647 | 0.460 |
| ETTh2 | **0.378** | 0.396 | 0.379 | 0.382 | **0.377** | 0.381 | 0.407 | 0.497 | 0.407 | 0.683 | 0.515 | 0.723 | 0.449 |
| ETTm1 | 0.378 | 0.391 | 0.410 | 0.388 | 0.389 | **0.371** | 0.410 | 0.406 | 0.400 | 0.495 | 0.407 | 0.481 | 0.452 |
| ETTm2 | **0.307** | 0.329 | 0.341 | 0.321 | 0.320 | **0.306** | 0.332 | 0.332 | 0.326 | 0.610 | 0.401 | 0.537 | 0.349 |

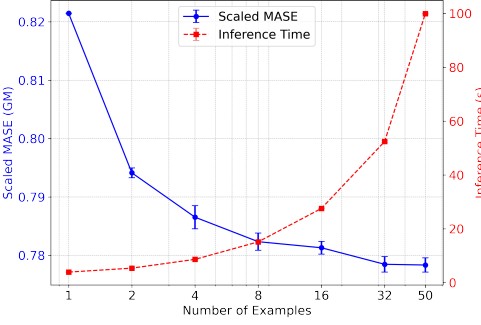

*Figure 6.* Scaled MASE (GM) vs number of in-context examples over the short context datasets in the OOD Benchmark. We also plot the total inference time for all the datasets as we vary the number of examples. All numbers are averaged over 5 runs with the corresponding one standard error.

the standard error bars are reported. We can see that the error decreases monotonically as the number of in-context examples are increased. At the same time, the total inference time increases, signifying a trade-off.

### 6.4.2. LONGER HISTORY

In this section, we compare the performance of TimesFM-ICF with a modified version of TimesFM (base) pretrained[4] and evaluated with a longer history $L = 2048$ which we will refer to as TimesFM (LH). We provide the aggregate scaled MASE on the OOD Benchmark in Table 2. We restrict TimesFM-ICF to use at most 5 in-context examples (of length 512 each), so it is a fair comparison.

| Dataset | TimesFM-ICF | TimesFM (LH) | TimesFM (base) |
| --- | --- | --- | --- |
| OOD Benchmark | **0.777** | 0.811 | 0.834 |

*Table 2.* Comparison with longer context TimesFM (LH) model, which has a maximum history of 2048. We report the Scaled MASE (GM) on the OOD Benchmark.

We can see that TimesFM (LH) yields a modest 2.4% improvement over TimesFM (base) (maximum history of 512) while TimesFM-ICF yields a 6.8% improvement.

---

[4]Pretraining performed in a manner similar to the latest version of the TimesFM Hugging Face repo.

This shows that our technique of in-context fine-tuning can be more effective than training a longer history model, especially when there is a mix of short-history and long-history time-series. This is because, for in-context fine-tuning, many short time-series can be packed as in-context examples inside the context, while for the case of usual long history training such time-series will just be padded and most of the context is wasted. As shown in the detailed results in Table 6, the long history model performs at par or very slightly better on longer datasets like ERCOT, but degrades on shorter datasets like CIF and Toursim.

## 7. Conclusions

In this paper, we introduce a methodology for in-context fine-tuning of a time-series foundation model for forecasting. In particular, we start with a base foundation model and adapt it to be able to effectively utilize, at inference time, not just the history of the target time-series for forecasting, but also in-context examples from related time-series. Our results show that in-context fine-tuning can lead to significantly better zero-shot performance on popular forecasting benchmarks compared to the base foundation model and state-of-the-art supervised models. Furthermore, it even matches the performance of a version of the base foundation model that is explicitly fine-tuned on the target domain.

While we have chosen a specific base time-series foundation model (TimesFM) for our in-context fine-tuning approach, it would be an interesting direction of future work to study these adaptations for other base foundation models. It would also be interesting to study better forms of relative positional encodings specifically designed for handling in-context examples and length generalization.

## Impact Statement

This paper presents work whose goal is to advance the field of Machine Learning. There are many potential societal consequences of our work, none which we feel must be specifically highlighted here.

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

# A. Appendix

## A.1. Illustrative Examples

We illustrate visually in Figure 7 how in-context examples can help disambiguate the prediction tasks, by plotting the actual forecasts from TimesFM-ICF with and without the in-context examples. In the left two figures, the history is not sufficiently informative for the model to make an accurate prediction. By providing in-context examples together with this short history (see the right two figures), however, the model is able to make a more accurate forecast.

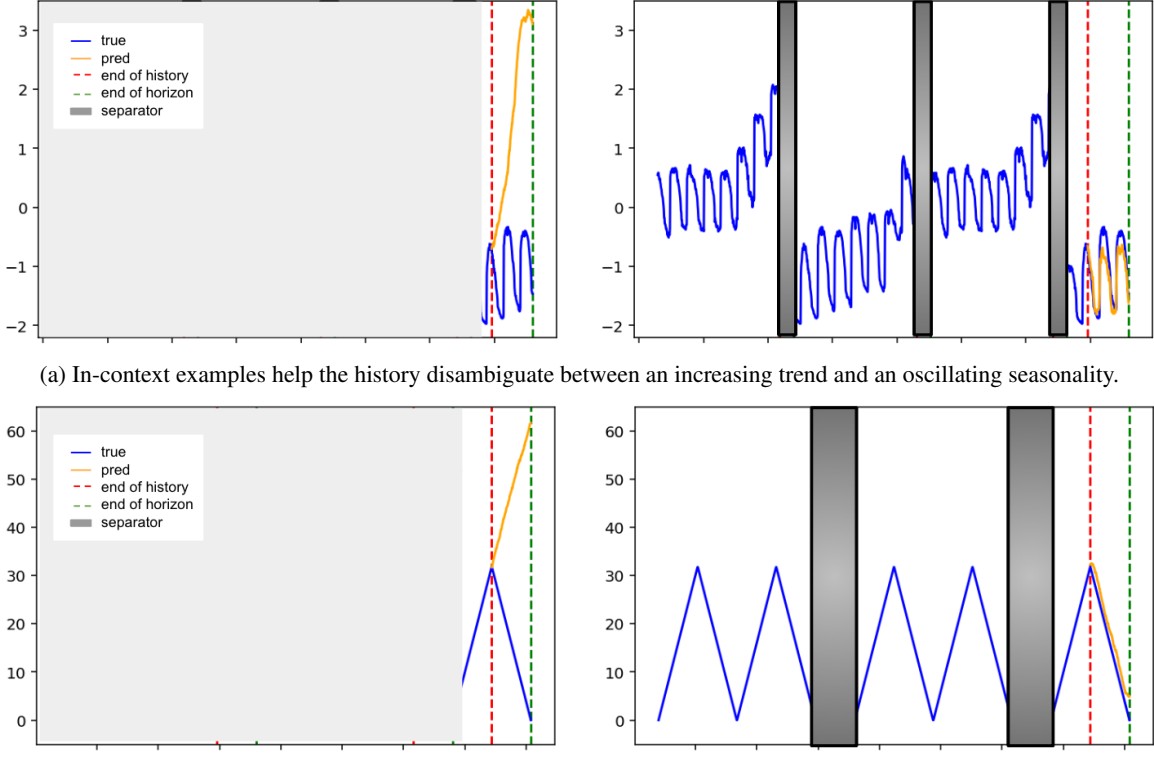

(a) In-context examples help the history disambiguate between an increasing trend and an oscillating seasonality.

(b) In-context examples help the history disambiguate between an increasing linear trend and a triangular wave.

*Figure* 7. Two illustrative examples on how in-context examples can help disambiguate the prediction tasks, that likely patterns based solely on the history can get proved or disproved by the patterns from the in-context examples.

In Figure 8, we plot the forecasts of TimesFM-ICF on the 5 time-series in the Monash Australian Electricity Demand dataset, operating our model in three modes: 0 in-context examples, 20 (random) in-context examples, and 50 in-context examples (5 of which are within-series examples). These three configurations have increasingly better MASE scores on this dataset (with MASE values 1, .9, and .8, respectively). The predictions visually appear to improve with the MASE values.

## A.2. Baselines on the OOD Benchmark

For the OOD Benchmark which is derived from the zero-shot Chronos benchmark, we borrow benchmark evaluation numbers from Table 10 in (Ansari et al., 2024) for our evaluations of the Chronos models, as well as that of LLMTime (Gruver et al., 2023), ForecastPFN (Dooley et al., 2024), Lag-Llama (Rasul et al., 2023), PatchTST (Nie et al., 2023), DeepAR (Salinas et al., 2020), WaveNet (van den Oord et al., 2016), TFT (Lim & Zohren, 2021) DLinear (Zeng et al., 2023), N-HiTS (Challu et al., 2023), N-BEATS (Oreshkin et al., 2020), GPT4TS (Zhou et al., 2023), SCUM (Petropoulos & Svetunkov, 2020) AutoETS, AutoARIMA, AutoTheta, Naïve, and Seasonal Naïve (Assimakopoulos & Nikolopoulos, 2000).

We omit the Moirai (Woo et al., 2024) evaluations from this benchmark, since this model's training data has more than 80% overlap with the benchmark.

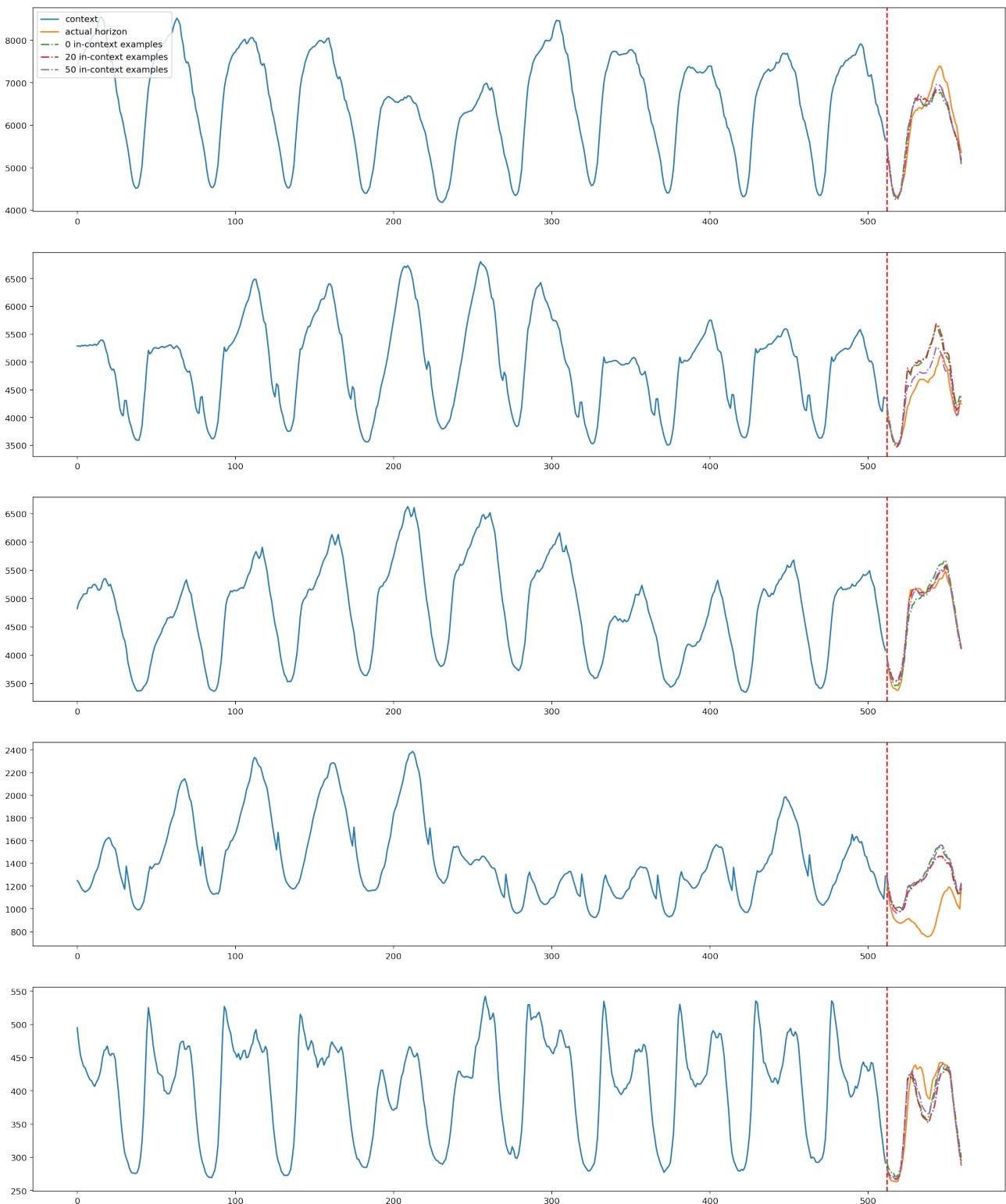

*Figure 8.* Visualization of TimesFM-ICF predictions on the Monash Australian Electricity dataset

We conduct the remaining evaluations ourselves, using the datasets available at this URL from the Chronos authors. We detail the datasets used in our evaluation in Table 4. To ensure our evaluations are zero-shot, we omit the M4 Quarterly, M4 Yearly[5], Traffic[6], and Weather[7] datasets from the Chronos zero-shot benchmark.

We give a detailed breakdown in Appendix A.6.

### A.3. Details of Pretraining TimesFM (base)

We start from the model architecture in Das et al. (2024) then create TimesFM (base) with 16 attention heads, 50 layers, an input patch length of 32 and output patch length of 128. The model dimension is set to 1280. We use the learning rate schedule in (Vaswani et al., 2017) with peak learning rate of $5e - 4$. The hidden dims of both the residual block and the FFN in the transformer layers are set as the same as model dimensions. We keep layer norm in transformer layers but not in the residual blocks. The pretraining datasets are detailed in Table 3.

The only difference between the model in Das et al. (2024) and our base model is that we use No Positional Encodings (NoPE) instead of the absolute positional encoding (Vaswani et al., 2017).

Based on the findings in Haviv et al. (2022), we create the pretrained TimesFM (base) checkpoint with NoPE, in contrast to the absolute positional encodings (Vaswani et al., 2017) used in the original TimesFM model. We note that we can achieve the same accuracy reported in the original TimesFM paper without using any positional encodings. Indeed it has been hypothesized in Haviv et al. (2022) that the presence of causal attention itself can encode positional information when there are more than one stacked transformer layers.

The advantages of NoPE for our continued pretraining are two fold: (i) NoPE models usually have better length generalization, which is particularly important here since we increase the prompt length by adding in-context examples to the context. (ii) If we use the original absolute positional encodings used in (Das et al., 2024), the meaning of these positional encodings in the base model would be different from their meaning during the continued pretraining with in-context examples. This could cause problems for the continued pretraining phase.

Empirically, NoPE leads to no loss in accuracy on validation during training, and works on par with other positional encodings that generalize length, e.g., FIRE (Li et al., 2024), see Figure 9.

### A.4. Details of TimesFM-FT: Fine-Tuning Per Dataset

On both the OOD Benchmark and the Long Horizon ETT, we also compare with TimesFM-FT which fine-tunes on the train split for every dataset and the forecasting on the corresponding test split. For all our fine-tuning runs, we use a batch size of 16 and (1) up to 10k iterations for the OOD Benchmark and (2) up to 100k for the Long Horizon ETT. We use a maximum learning rate of 1e-3, with 500-step linear warm-up and exponential decay. Note that this means that the fine-tuned model will see many more training examples than the in-context examples given to our model. For the fine-tuning runs, we use the same decoder only loss function that was used in the original pretraining of TimesFM (base). We do two kinds of fine-tuning:

- *Full:* All weights in the model are updated during fine-tuning.

- *Linear Probing (LP):* We hold the transformer weights fixed and only update the parameters in the input and output residual blocks.

### A.5. Additional Details of TimesFM-ICF

We continue to train TimesFM-ICF model from TimesFM (base). Therefore, most of the parameters in the model remain the same. Here, are the key training details that are unique to TimesFM-ICF:

- *Separator Token:* We have a trainable separator token that is also updated during the continued pretraining. The token is nothing but a learnt embedding whose dimension is equal to the model dimension i.e. 1280 in our case.

---

[5] https://github.com/Mcompetitions/M4-methods
[6] https://zenodo.org/record/4656132
[7] https://zenodo.org/record/4654822

- *Number of Examples:* We use a maximum of $n = 50$ in-context examples for each context during training.

- *Padding:* In short datasets like M4 yearly and quarterly, each time-series might have number of time-points much less than $T = 640$. Sometimes the number of time-points are even less than our input patch length $p = 32$. For such cases, a whole time-series can fit into one of the $n$ examples and they are preprocessed in the following manner:

  - If the length of the time-series $l$ is less than $p$, we left pad with $k$ padding time-points such that $p < k + l < 2p$. This is because we want the decoder only model to predict something meaningful for the second patch after seeing the first patch and if not, is penalized by the loss on the second patch. If the $l > p$, we do not need to perform this left padding.
  - Lastly, we right pad such that the length of the total padded example is $T = 640$.
  - Note that the last patch in such examples would be padded from the right, i.e., they will have real time-series values for the first few points and padding for the rest. We make sure that such incomplete from the right patches are not attended by subsequent tokens belonging to examples coming after.

The continued pretraining datasets are detailed in Table 3.

### A.6. OOD Benchmark Detailed Results

We give a detailed breakdown of the zero-shot evaluations on the datasets from Table 4 (displayed in Figure 5) in Table 6 with additional baselines as mentioned in Appendix A.2. We report the mean absolute scaled error (MASE) (Hyndman & Koehler, 2006), which, for a given time-series $\mathbf{y}_{1:L+H} = (y_1, \ldots, y_L, \ldots, y_{L+H})$ with context $L$ and horizon $H$ and seasonality parameter $S$, together with a predicted time-series $\widehat{\mathbf{y}}_{L+1:L+H} = (\hat{y}_{L+1}, \ldots, \hat{y}_{L+H})$, is defined as the mean absolute error for the forecast normalized by the seasonal naïve forecast error on the context:

$$\text{MASE}(\widehat{\mathbf{y}}_{L+1:L+H}, \mathbf{y}_{1:L+H}) = \frac{L - S}{H} \frac{\sum_{t=L+1}^{L+H} |\hat{y}_t - y_t|}{\sum_{t=1}^{L-S} |y_t - y_{t+S}|}.$$

Each evaluation of TimesFM-ICF is averaged over 10 random seeds, where the randomness corresponds to the random selection of in-context examples used to make predictions. The reported MASE numbers are averaged over the five evaluations of the dataset. The confidence intervals in Table 6 correspond to one standard deviation of the 5 evaluations, averaged over the dataset.

Since TimesFM (base) supports probabilistic forecasting, in Table 7, we additionally report the weighted quantile loss (WQL), which averages the quantile loss $\text{QL}_\alpha$ of (Koenker & Hallock, 2001) over quantiles $\alpha \in \{0.1, 0.2, \ldots, 0.9\}$. The quantile loss is defined as follows: given quantile $\alpha \in (0, 1)$ and predicted quantiles $\mathbf{q}^{(\alpha)} = (q_{L+1}^{(\alpha)}, \ldots, q_{L+H}^{(\alpha)})$ for a time-series $\mathbf{y}$ with context $L$ and horizon $H$, the quantile loss at level $\alpha$ of an observation $y_t$ is:

$$\text{QL}_\alpha(q_t^\alpha, y_t) = \begin{cases} \alpha(y_t - q_t^{(\alpha)}) & \text{if } y_t > q_t^{(\alpha)} \\ (1 - \alpha)(q_t^{(\alpha)} - y_t) & \text{otherwise} \end{cases}$$

The weighted quantile loss over a dataset $\mathcal{D}$, where each $\mathbf{y}_i \in \mathcal{D}$ has a predicted quantile $\mathbf{q}_i^{(\alpha)}$, is given by:

$$\text{WQL}_\alpha = \frac{2 \sum_{i \in |\mathcal{D}|} \sum_{t=L+1}^{L+H} \text{QL}_\alpha(q_{i,t}^\alpha, y_{i,t})}{\sum_{i \in |\mathcal{D}|} \sum_{t=L+1}^{L+H} |y_{i,t}|}$$

Then, the weighted quantile loss is the average of $\text{WQL}_\alpha$ over $\alpha \in \{0.1, 0.2, \ldots, 0.9\}$.

Note that, in addition to the detailed results in Tables 6 and 7, we additionally report the geometric mean of the MASE on each dataset, normalized by the MASE of the seasonal naïve baseline. This geometric mean is then averaged over the 5 evaluation runs. We report both the geometric mean of the 23 datasets from the Chronos zero-shot benchmark for which we are zero-shot (see Table 4), i.e., that were not used during training of TimesFM-ICF or the base model TimesFM (base), as "Geometric Mean (ZS)". We also report the the geometric mean over all 27 datasets from the Chronos zero-shot benchmark as "Geometric Mean (All)".

*Table 3.* List of datasets included in pretraining TimesFM (base). All datasets except the Wiki datasets are also repurposed for continued pretraining TimesFM-ICF with in-context examples. The datasets labeled LOTSA are obtained from the LOTSA collection (Woo et al., 2024). For continued pretraining, the Wiki dataset can be clustered into groups of related articles, and the time-series for each group could be deemed to form a separate dataset. But we leave such preprocessing of the Wiki dataset for future work and leave these datasets out of our continued pretraining.

| Dataset | Granularity | # Time series |
|---|---|---|
| Synthetic | | 3,000,000 |
| Electricity | Hourly | 321 |
| Traffic | Hourly | 862 |
| Weather (Zhou et al., 2021) | 10 Min | 42 |
| Favorita Sales | Daily | 111,840 |
| LibCity (Jiang et al., 2023) | 15 Min | 6,159 |
| M4 hourly | Hourly | 414 |
| M4 daily | Daily | 4,227 |
| M4 monthly | Monthly | 48,000 |
| M4 quarterly | Quarterly | 24,000 |
| M4 yearly | Yearly | 22,739 |
| Wiki hourly | Hourly | 5,608,693 |
| Wiki daily | Daily | 68,448,204 |
| Wiki weekly | Weekly | 66,579,850 |
| Wiki monthly | Monthly | 63,151,306 |
| Trends hourly | Hourly | 22,435 |
| Trends daily | Daily | 22,435 |
| Trends weekly | Weekly | 22,435 |
| Trends monthly | Monthly | 22,435 |
| LOTSA Azure VM Traces | 5 Min | 7,179,194 |
| LOTSA Residential LoadPower | 1 Min | 6,861,408 |
| LOTSA Borg Cluster Data | 5 Min | 6,651,848 |
| LOTSA Residential PvPower | 1 Min | 5,890,848 |
| LOTSA QTraffic | 15 Min | 4,334,208 |
| LOTSA London Smart Meters | 30 Min | 2,681,312 |
| LOTSA Taxi | 30 Min | 2,175,488 |
| LOTSA Solar Power | 4 Sec | 115,584 |
| LOTSA Wind Power | 4 Sec | 115,584 |
| LOTSA Kdd2022 | 10 Min | 77184 |
| LOTSA Largest | 5 Min | 91,005,824 |
| LOTSA Era5 | Hourly | 728,078,400 |
| LOTSA Buildings | Hourly | 22,937,600 |
| LOTSA Cmip6 | Daily | 892,769,472 |
| LOTSA China Air Quality | Hourly | 593,532 |
| LOTSA Beijing Air Quality | Hourly | 76,032 |
| LOTSA Subseasonal | Daily | 165,504 |
| LOTSA Kaggle WebTraffic Weekly | Weekly | 4,642,016 |
| LOTSA CdC Flu | Weekly | 9,472 |
| LOTSA Godaddy | Monthly | 200,640 |

## A.7. Detailed Timing on the OOD Benchmark

We provide the comparison between TimesFM-ICF and TimesFM-FT regarding their benchmark timing and accuracy in Table 8. For a fair comparison we calculate the time here assuming that fine-tuning and inference tasks on all datasets are completed in a one-by-one sequential manner. Fine-tuning on each dataset requires more time than applying in-context examples while does not always guarantee performance over TimesFM-ICF.

*Table 4.* Details of the OOD Benchmark: We report our results on the 23 datasets from the Chronos zero-shot benchmark that were not used during training of TimesFM-ICF or the base model TimesFM (base).

| Dataset | Category | Frequency | Num TS | MinLen | AvgLen | MaxLen | Horizon |
|---|---|---|---|---|---|---|---|
| Australian Electricity | energy | 30min | 5 | 230736 | 231052 | 232272 | 48 |
| Car Parts | retail | 1M | 2674 | 51 | 51 | 51 | 12 |
| CIF 2016 | banking | 1M | 72 | 28 | 98 | 120 | 12 |
| Covid Deaths | healthcare | 1D | 266 | 212 | 212 | 212 | 30 |
| Dominick | retail | 1D | 100014 | 201 | 296 | 399 | 8 |
| ERCOT Load | energy | 1H | 8 | 154854 | 154854 | 154854 | 24 |
| ETT (15 Min., Last Window) | energy | 15min | 14 | 69680 | 69680 | 69680 | 24 |
| ETT (Hourly, Last Window) | energy | 1H | 14 | 17420 | 17420 | 17420 | 24 |
| Exchange Rate | finance | 1B | 8 | 7588 | 7588 | 7588 | 30 |
| FRED-MD | economics | 1M | 107 | 728 | 728 | 728 | 12 |
| Hospital | healthcare | 1M | 767 | 84 | 84 | 84 | 12 |
| M1 (Monthly) | various | 1M | 617 | 48 | 90 | 150 | 18 |
| M1 (Quarterly) | various | 3M | 203 | 18 | 48 | 114 | 8 |
| M1 (Yearly) | various | 1Y | 181 | 15 | 24 | 58 | 6 |
| M3 (Monthly) | various | 1M | 1428 | 66 | 117 | 144 | 18 |
| M3 (Quarterly) | various | 3M | 756 | 24 | 48 | 72 | 8 |
| M3 (Yearly) | various | 1Y | 645 | 20 | 28 | 47 | 6 |
| M5 | retail | 1D | 30490 | 124 | 1562 | 1969 | 28 |
| NN5 (Daily) | finance | 1D | 111 | 791 | 791 | 791 | 56 |
| NN5 (Weekly) | finance | 1W | 111 | 113 | 113 | 113 | 8 |
| Tourism (Monthly) | various | 1M | 366 | 91 | 298 | 333 | 24 |
| Tourism (Quarterly) | various | 1Q | 427 | 30 | 99 | 130 | 8 |
| Tourism (Yearly) | various | 1Y | 518 | 11 | 24 | 47 | 4 |

All timing is reported on TPUv5e with 4 tensor cores.

## A.8. ETT Rolling Window

We provide the detailed results of the mean absolute error (MAE) of TimesFM-ICF against other supervised and zero-shot methods on ETT Rolling Window in Table 9. The MAE of a prediction $\widehat{\mathbf{y}}_{L+1:L+H}$ of a time-series $\mathbf{y}_{1:L+H}$ is:

$$\text{MAE}(\widehat{\mathbf{y}}_{L+1:L+H}, \mathbf{y}_{1:L+H}) = \frac{1}{H}\|\widehat{\mathbf{y}}_{L+1:L+H} - \mathbf{y}_{L+1:L+H}\|_1.$$

Note that, in addition to the baselines Table 1, we additionally evaluate against the Moirai (Large) model operating in multivariate forecasting mode. We refer to Moirai operating in this mode as Moirai-MV. We provide Moirai-MV with a total of 50 multivariate features – 21 (the maximum possible number for these datasets) from the time-series' history, and 29 features from randomly selected time series. We select this for several reasons: (i) by selecting the largest possible context history, we provide the model with the largest possible number of relevant features, (ii) according to (Woo et al., 2024), Moirai is trained to accommodate at least 50 multivariate features, and (iii) our model is trained to accommodate a maximum of 50 in-context examples. We choose each multivariate feature to have the same length (512) since both TimesFM-ICF and Moirai-MV is trained to accommodate such example lengths. Note that we provide exactly the same features/in-context examples to Moirai-MV and to TimesFM-ICF , for ease of comparison. Moirai-MV's performance degrades on these datasets with the additional multivariate features provided.

We remark, however, that the performance degradation of the Moirai-MV model may be explained by a subtle distinction between in-context examples and multivariate features. All multivariate features used when training Moirai are assumed to be temporally aligned. By contrast, the in-context examples provided during our evaluations are from different time-windows. Thus, by providing in-context examples as multivariate features for Moirai-MV, we are evaluating the model outside of its intended use. Nonetheless, we include this result in the appendix (with this caveat) to demonstrate that naïvely concatenating related time-series as multivariate features may not work out-of-the-box.

*Table 5.* Part of the OOD Benchmark which only has room for one in-context example per time-series. The is used for the ablation study in Section 6.4.1. We exclude Dominick as evaluation takes a significant amount of time on that dataset especially when averaged over multiple runs.

| Dataset | Category | Frequency | Num TS | MinLen | AvgLen | MaxLen | Horizon |
|---|---|---|---|---|---|---|---|
| Car Parts | retail | 1M | 2674 | 51 | 51 | 51 | 12 |
| CIF 2016 | banking | 1M | 72 | 28 | 98 | 120 | 12 |
| Covid Deaths | healthcare | 1D | 266 | 212 | 212 | 212 | 30 |
| FRED-MD | economics | 1M | 107 | 728 | 728 | 728 | 12 |
| Hospital | healthcare | 1M | 767 | 84 | 84 | 84 | 12 |
| M1 (Monthly) | various | 1M | 617 | 48 | 90 | 150 | 18 |
| M1 (Quarterly) | various | 3M | 203 | 18 | 48 | 114 | 8 |
| M1 (Yearly) | various | 1Y | 181 | 15 | 24 | 58 | 6 |
| M3 (Monthly) | various | 1M | 1428 | 66 | 117 | 144 | 18 |
| M3 (Quarterly) | various | 3M | 756 | 24 | 48 | 72 | 8 |
| M3 (Yearly) | various | 1Y | 645 | 20 | 28 | 47 | 6 |
| M5 | retail | 1D | 30490 | 124 | 1562 | 1969 | 28 |
| NN5 (Daily) | finance | 1D | 111 | 791 | 791 | 791 | 56 |
| NN5 (Weekly) | finance | 1W | 111 | 113 | 113 | 113 | 8 |
| Tourism (Monthly) | various | 1M | 366 | 91 | 298 | 333 | 24 |
| Tourism (Quarterly) | various | 1Q | 427 | 30 | 99 | 130 | 8 |
| Tourism (Yearly) | various | 1Y | 518 | 11 | 24 | 47 | 4 |

*Table 6.* OOD Benchmark (MASE)

| | Pretrained Models (Few Shot) | | | | | | | | | | Pretrained Models (Other) / Pretrained Models (Zero Shot) | | Task Specific Models | | | | | | | | | | | | Local Models | |
|---|---|---|---|---|---|---|---|---|---|---|---|---|---|---|---|---|---|---|---|---|---|---|---|---|---|---|
| Model | TimesFM-ICL | TimesFM (Base) | TimesFM-LH | Chronos-T5 (Large) | Chronos-T5 (Base) | Chronos-T5 (Small) | Chronos-T5 (Mini) | Chronos-GPT2 | LLMTime | ForecastPFN | Lag-Llama | TimesFM-FT | PatchTST | DeepAR | WaveNet | TFT | DLinear | N-HiTS | N-BEATS | GPT4TS | SCUM | AutoETS | AutoTheta | AutoARIMA | Seasonal Naive | Naive |
| Australian Electricity | 0.805 ± 0.010 | 1.529 | 1.313 | 1.333 | 1.319 | 1.399 | 1.114 | 1.310 | 1.186 | 2.158 | 1.635 | **0.753** | 0.871 | 1.473 | 0.997 | 0.810 | 1.278 | 0.794 | 0.828 | 1.161 | 1.427 | 2.391 | 0.897 | 1.393 | 1.253 | 2.362 |
| Car Parts | 0.868 ± 0.0002 | 0.922 | 0.922 | 0.906 | 0.899 | 0.887 | 0.891 | 0.883 | - | 2.657 | 0.816 | 0.867 | 0.803 | **0.798** | 0.817 | 0.799 | 0.879 | 0.803 | 0.891 | 1.157 | 1.185 | 1.229 | - | - | 1.201 | - |
| CIF 2016 | 0.377 ± 0.003 | 0.948 | 0.947 | 0.986 | 0.981 | 0.989 | 1.051 | 1.046 | 1.384 | 3.588 | 2.235 | **0.860** | 1.537 | 1.363 | 1.309 | 1.553 | 1.145 | 1.389 | 1.440 | 0.960 | 0.907 | 1.157 | 1.002 | 1.006 | 1.289 | 1.263 |
| Covid Deaths | 44.279 ± 0.077 | 47.399 | 47.391 | 42.550 | 42.687 | 42.670 | 43.621 | 48.215 | 32.143 | 91.515 | 78.456 | 39.904 | 36.465 | 38.203 | 102.457 | **30.635** | 40.418 | 31.771 | 31.730 | 75.909 | 33.595 | 38.114 | 45.407 | 31.705 | 46.912 | 46.912 |
| Dominick | 0.900 ± 0.0002 | 0.935 | 0.935 | 0.818 | 0.816 | 0.819 | 0.833 | 0.820 | - | 3.274 | 1.250 | 0.835 | 0.867 | 0.851 | 0.812 | 0.800 | 0.880 | **0.782** | 0.782 | 1.813 | 0.891 | 0.885 | 1.016 | - | 0.871 | 0.871 |
| ERCOT Load | 0.669 ± 0.013 | 0.633 | 0.690 | 0.617 | **0.550** | 0.573 | 0.588 | 0.561 | 1.319 | 3.975 | 0.834 | 0.644 | 0.553 | 1.197 | 0.780 | 0.690 | 0.651 | 0.615 | 0.648 | 0.558 | 1.308 | 2.826 | 1.306 | 1.284 | 0.761 | 4.234 |
| ETT (15 Min.) | 0.653 ± 0.004 | 0.767 | 0.615 | 0.741 | 0.739 | 0.710 | 0.792 | 0.796 | 1.042 | 1.138 | 0.967 | 0.742 | 0.652 | 0.874 | 1.339 | 0.962 | 0.724 | 0.643 | 0.659 | **0.574** | 0.673 | 1.183 | 0.583 | 0.879 | 1.169 | 1.164 |
| ETT (Hourly) | 0.801 ± 0.003 | 0.838 | 0.853 | 0.735 | 0.789 | 0.789 | 0.797 | 0.768 | 1.232 | 1.833 | 1.002 | 0.798 | 0.729 | 0.814 | 1.509 | 0.875 | **0.695** | 0.811 | 0.782 | 0.768 | 0.850 | 1.139 | 0.900 | 0.977 | 0.932 | 1.651 |
| Exchange Rate | 1.632 ± 0.036 | 2.113 | 1.970 | 2.375 | 2.433 | 2.252 | 2.030 | 2.335 | 1.743 | 7.583 | 3.087 | 1.871 | 1.540 | 1.615 | 3.105 | 2.361 | **1.459** | 2.041 | 2.149 | 2.709 | 1.749 | 1.643 | 1.648 | 1.882 | 1.740 | 1.874 |
| FRED-MD | 0.610 ± 0.003 | 0.653 | 0.492 | 0.500 | 0.486 | 0.496 | 0.483 | **0.468** | 0.513 | 2.621 | 2.283 | 0.592 | 0.745 | 0.621 | 0.840 | 0.929 | 0.713 | 0.696 | 0.635 | 0.693 | 0.492 | 0.544 | 0.566 | 0.473 | 1.101 | 0.622 |
| Hospital | 0.752 ± 0.0004 | 0.780 | 0.780 | 0.810 | 0.810 | 0.815 | 0.817 | 0.831 | 0.861 | 1.775 | 0.939 | 0.763 | 0.839 | 0.857 | 0.799 | 0.760 | 0.781 | 0.760 | 0.761 | 0.829 | **0.748** | 0.760 | 0.761 | 0.820 | 0.921 | 0.968 |
| M1 (Monthly) | **1.013 ± 0.0015** | 1.052 | 1.052 | 1.090 | 1.117 | 1.169 | 1.174 | 1.182 | 1.415 | 2.172 | 1.875 | 1.083 | 1.208 | 1.122 | 1.266 | 1.326 | 1.369 | 1.333 | 1.236 | 1.198 | 1.023 | 1.099 | 1.153 | - | 1.314 | 1.468 |
| M1 (Quarterly) | 1.652 ± 0.005 | 1.709 | 1.709 | 1.713 | 1.739 | 1.764 | 1.785 | 1.785 | 1.802 | 9.931 | 3.036 | 1.704 | 1.920 | 1.741 | 1.904 | 2.144 | 1.943 | 2.061 | 2.043 | 1.958 | **1.602** | 1.710 | 1.683 | 1.770 | 2.078 | 1.952 |
| M1 (Yearly) | 4.299 ± 0.008 | 3.591 | 3.592 | 4.301 | 4.624 | 4.659 | 4.958 | 4.751 | 4.077 | 23.089 | 7.149 | **3.284** | 4.042 | 3.685 | 4.727 | 11.565 | 5.568 | 6.212 | 3.675 | 3.571 | 4.110 | 3.697 | 3.870 | - | 4.894 | 4.894 |
| M3 (Monthly) | 0.837 ± 0.001 | 0.832 | 0.832 | 0.857 | 0.868 | 0.885 | 0.900 | 0.930 | 0.996 | 2.240 | 1.846 | **0.811** | 1.225 | 0.943 | 0.950 | 0.916 | 1.161 | 0.899 | 0.883 | 0.950 | 0.827 | 0.869 | 0.861 | 0.933 | 1.146 | 1.175 |
| M3 (Quarterly) | 1.140 ± 0.002 | 1.201 | 1.202 | 1.181 | 1.199 | 1.256 | 1.289 | 1.241 | 1.450 | 10.176 | 2.886 | 1.161 | 1.264 | 1.209 | 1.257 | 1.160 | 1.572 | 1.202 | 1.147 | 1.448 | 1.135 | **1.125** | 1.130 | 1.419 | 1.425 | 1.464 |
| M3 (Yearly) | 2.779 ± 0.004 | 2.818 | 2.817 | 3.106 | 3.209 | 3.276 | 3.385 | 3.158 | 3.140 | 18.728 | 5.114 | 2.796 | 2.949 | 2.827 | 3.026 | 2.860 | 3.435 | 3.432 | 3.547 | 3.418 | 2.703 | 2.696 | **2.613** | 3.165 | 3.172 | 3.172 |
| M4 (Quarterly) | 1.220 ± 0.001 | **0.966** | 0.966 | 1.216 | 1.231 | 1.246 | 1.271 | 1.312 | - | 6.927 | 2.663 | 0.970 | 1.150 | 1.254 | 1.241 | 1.248 | 1.229 | 1.157 | 1.129 | 1.215 | 1.145 | 1.188 | 1.193 | 1.276 | 1.602 | 1.477 |
| M4 (Yearly) | 3.311 ± 0.002 | 2.549 | **2.549** | 3.606 | 3.678 | 3.651 | 3.743 | 3.933 | - | - | 5.866 | 2.554 | 3.072 | 3.178 | 3.221 | 3.119 | 3.295 | - | - | 3.374 | 3.013 | 3.374 | 3.124 | 3.730 | 3.974 | 3.974 |
| M5 | 0.924 ± 0.00003 | 0.916 | 0.920 | 0.944 | 0.939 | 0.940 | 0.944 | 0.999 | - | 1.530 | 0.965 | **0.904** | 0.919 | 0.956 | 0.959 | 0.909 | 1.027 | 0.917 | 0.917 | 0.935 | 1.096 | 1.101 | 1.100 | 1.057 | 1.399 | 1.399 |
| NN5 (Daily) | 0.578 ± 0.001 | 0.612 | 0.603 | 0.573 | 0.585 | 0.615 | 0.642 | 0.601 | 0.953 | 1.375 | 0.992 | 0.572 | 0.575 | 0.585 | **0.556** | 0.604 | 0.571 | 0.571 | 0.720 | 1.052 | 1.039 | 1.073 | 1.214 | - | 1.292 | 1.292 |
| NN5 (Weekly) | 0.919 ± 0.003 | **0.865** | 0.865 | 0.940 | 0.938 | 0.944 | 0.947 | 0.963 | 0.968 | 1.349 | 1.141 | 0.933 | 0.877 | 0.920 | 1.034 | 0.896 | 0.966 | 0.919 | 1.014 | 1.268 | 0.974 | 0.978 | 0.984 | 0.995 | 1.063 | 1.063 |
| Tourism (Monthly) | 1.487 ± 0.002 | 1.623 | 1.623 | 1.761 | 1.828 | 1.900 | 1.950 | 1.783 | 2.139 | 4.348 | 3.030 | 1.516 | 1.572 | 1.529 | 1.629 | 1.686 | 1.551 | 1.514 | 1.486 | 1.573 | **1.441** | 1.497 | 1.680 | 1.573 | 1.631 | 1.591 |
| Tourism (Quarterly) | 1.708 ± 0.0041 | 1.799 | 1.798 | 1.677 | 1.717 | 1.730 | 1.829 | 1.828 | 1.769 | 5.995 | 3.695 | 1.776 | 1.723 | 1.586 | 1.760 | 1.729 | 1.690 | 1.585 | 1.618 | 1.750 | **1.501** | 1.590 | 1.658 | 1.661 | 1.699 | 3.633 |
| Tourism (Yearly) | 3.224 ± 0.006 | 3.496 | 3.496 | 3.755 | 3.900 | 4.048 | 3.662 | 3.309 | 12.093 | 3.755 | 3.565 | **3.138** | 3.702 | 4.130 | 3.847 | 3.406 | 3.448 | 3.564 | - | 3.276 | 3.138 | 3.078 | 4.043 | - | 3.552 | 3.552 |
| Traffic | 0.837 ± 0.001 | **0.573** | 0.730 | 0.804 | 0.828 | 0.837 | 0.850 | 0.818 | 0.973 | 1.909 | 0.829 | 0.592 | 0.790 | 0.737 | 0.797 | 0.880 | 0.821 | 0.927 | 0.968 | 0.787 | - | 1.685 | 1.794 | - | 1.077 | 2.052 |
| Weather | 0.824 ± 0.001 | 0.871 | 0.830 | 0.822 | 0.824 | 0.836 | 0.853 | 0.858 | - | 2.003 | 1.001 | **0.782** | 0.860 | 0.911 | 0.945 | 0.913 | 0.907 | 0.910 | 0.888 | 0.972 | 0.933 | 1.079 | 0.991 | 0.907 | 1.004 | 1.004 |
| Geometric Mean (ZS) | 0.777 ± 0.003 | 0.834 | 0.812 | 0.827 | 0.834 | 0.844 | 0.852 | 0.852 | 0.960 | 2.543 | 1.313 | **0.776** | 0.818 | 0.852 | 0.977 | 0.851 | 0.905 | 0.824 | 0.830 | 0.909 | 0.837 | 0.943 | 0.856 | 0.908 | 1.000 | 1.195 |
| Geometric Mean (All) | 0.778 ± 0.002 | 0.805 | 0.792 | 0.824 | 0.832 | 0.841 | 0.850 | 0.852 | 0.962 | 2.450 | 1.291 | **0.753** | 0.810 | 0.843 | 0.951 | 0.847 | 0.894 | 0.830 | 0.835 | 0.896 | 0.838 | 0.953 | 0.875 | 0.908 | 1.000 | 1.189 |

## A.9. Selecting In-Context Examples

A practical consideration for in-context fine-tuning is the question of how to choose the (up to 50) related in-context examples to put in the context window (at both training and inference time). While our in-context fine-tuning methodology is compatible with any algorithm for in-context example selection, in this paper we use a very simple strategy of choosing for each time-series forecasting target a combination of a) a few consecutive examples chosen from the immediate history of the time-series to be forecasted, and b) many examples chosen at random (across timestamp and time-series) from the past history of other time-series in the same dataset. We also tried a more sophisticated approach that uses dynamic time warping (DTW) (Serra & Arcos, 2014; Salvador & Chan, 2007) to select the top 20% most "similar" time series to the time-series of interest, and restricting the random examples from within those time-series. In Figure 10 we perform a lightweight ablation study to understand the effects of these approaches on the performance of in-context fine-tuning on the OOD Benchmark, and observe very minor differences in performance among the example selection approaches. Choosing 5 examples from the immediate history, and the remaining 45 examples at random seemed to outperform the other three approaches (and is indeed the approach we use for TimesFM-ICF in the OOD Benchmark results).

As a simple test how our simple choice of in-context example selection could be improved, we performed a grid-search over various splits of in-series and random in-context examples as follows: First, we constructed a validation dataset from the training portion of a subset of the Monash datasets (specifically: weather, traffic, australian electricity, ercot, ETTm, and ETTh). We chose these datasets because they contained many training examples long enough to construct up to 20

*Table 7.* OOD Benchmark (WQL)

| Category | Pretrained Models (Few Shot) | | | | | | | | | Pretrained Models (Zero Shot) | Pretrained Models (Other) | Task Specific Models | | | | | | | | | | | Local Models | |
|---|---|---|---|---|---|---|---|---|---|---|---|---|---|---|---|---|---|---|---|---|---|---|---|---|
| Model | TimesFM-ICL | TimesFM (Base) | TimesFM-LH | Chronos-T5 (Large) | Chronos-T5 (Base) | Chronos-T5 (Small) | Chronos-T5 (Mini) | Chronos-GPT2 | LLMTime | Lag-Llama | TimesFM-FT | PatchTST | DeepAR | WaveNet | TFT | DLinear | N-HiTS | N-BEATS | SCUM | AutoETS | AutoTheta | AutoARIMA | Seasonal Naive | Naive |
| Australian Electricity | 0.037 | 0.078 | 0.067 | 0.067 | 0.075 | 0.074 | 0.063 | 0.078 | 0.069 | 0.097 | **0.833** | 0.037 | 0.087 | 0.052 | 0.036 | 0.066 | 0.034 | 0.038 | 0.070 | 0.125 | 0.055 | 0.073 | 0.084 | 0.159 |
| CIF 2016 | 0.013 | 0.049 | 0.049 | 0.014 | 0.013 | 0.015 | 0.013 | 0.015 | 0.014 | 0.041 | 0.046 | 0.140 | 0.136 | 0.086 | 0.011 | 0.033 | 0.032 | 0.039 | 0.024 | 0.039 | 0.027 | 0.017 | 0.015 | **0.009** |
| Car Parts | 0.996 | 1.046 | 1.046 | 1.060 | 1.057 | 1.029 | 1.024 | 1.028 | - | 1.011 | 0.983 | 0.998 | 0.967 | 0.941 | **0.871** | 1.119 | 0.880 | 0.877 | 1.283 | 1.309 | 1.337 | - | 1.600 | - |
| Covid Deaths | 0.073 | 0.070 | 0.070 | 0.045 | 0.048 | 0.059 | 0.084 | 0.079 | 0.032 | 0.276 | 0.043 | 0.065 | 0.108 | 0.918 | 0.034 | 0.077 | 0.038 | 0.056 | 0.037 | 0.064 | 0.094 | **0.029** | 0.133 | 0.133 |
| Dominick | 0.355 | 0.371 | 0.371 | 0.332 | 0.332 | 0.333 | 0.338 | 0.346 | 0.336 | - | 0.443 | 0.327 | 0.345 | 0.364 | 0.327 | 0.320 | 0.435 | 0.313 | **0.312** | 0.439 | 0.483 | 0.485 | - | 0.453 | 0.453 |
| ERCOT Load | 0.021 | 0.023 | 0.021 | 0.019 | **0.016** | 0.018 | 0.018 | 0.017 | 0.053 | 0.033 | 0.022 | 0.017 | 0.032 | 0.024 | 0.023 | 0.023 | 0.020 | 0.020 | 0.050 | 0.122 | 0.041 | 0.052 | 0.037 | 0.181 |
| ETT (15 Min.) | 0.056 | 0.069 | 0.051 | 0.068 | 0.069 | 0.064 | 0.072 | 0.073 | 0.088 | 0.080 | 0.060 | 0.054 | 0.069 | 0.113 | 0.075 | 0.071 | **0.051** | 0.053 | 0.061 | 0.095 | 0.079 | 0.073 | 0.141 | 0.121 |
| ETT (Hourly) | 0.080 | 0.083 | 0.085 | 0.073 | 0.081 | 0.080 | 0.085 | 0.080 | 0.122 | 0.106 | 0.084 | **0.071** | 0.081 | 0.142 | 0.082 | 0.076 | 0.081 | 0.074 | 0.087 | 0.132 | 0.133 | 0.105 | 0.122 | 0.202 |
| Exchange Rate | **0.008** | 0.012 | 0.015 | 0.013 | 0.014 | 0.013 | 0.012 | 0.013 | 0.015 | 0.011 | 0.010 | 0.010 | 0.009 | 0.016 | 0.011 | 0.008 | 0.010 | 0.011 | 0.011 | 0.010 | 0.010 | 0.011 | 0.013 | 0.015 |
| FRED-MD | 0.027 | 0.040 | 0.029 | 0.020 | 0.022 | **0.017** | **0.017** | 0.022 | 0.041 | 0.389 | 0.043 | 0.042 | 0.043 | 0.058 | 0.112 | 0.069 | 0.057 | 0.061 | 0.059 | 0.055 | 0.057 | 0.056 | 0.122 | 0.064 |
| Hospital | 0.051 | 0.054 | 0.054 | 0.056 | 0.056 | 0.057 | 0.058 | 0.057 | 0.066 | 0.093 | 0.051 | 0.070 | 0.056 | 0.064 | 0.053 | 0.089 | 0.052 | **0.050** | 0.052 | 0.053 | 0.055 | 0.058 | 0.073 | 0.087 |
| M1 (Monthly) | 0.148 | 0.130 | 0.130 | 0.130 | **0.128** | 0.139 | 0.138 | 0.131 | 0.181 | 0.196 | 0.145 | 0.165 | 0.150 | 0.150 | 0.175 | 0.189 | 0.189 | 0.187 | 0.162 | 0.162 | 0.159 | 0.146 | 0.191 | 0.258 |
| M1 (Quarterly) | 0.087 | 0.113 | 0.113 | 0.107 | 0.105 | 0.103 | 0.103 | 0.116 | 0.115 | 0.141 | 0.093 | **0.078** | 0.089 | 0.094 | 0.122 | 0.079 | 0.111 | 0.085 | 0.083 | 0.083 | 0.082 | 0.091 | 0.150 | 0.130 |
| M1 (Yearly) | 0.149 | 0.145 | 0.145 | 0.183 | 0.181 | 0.172 | 0.179 | 0.204 | 0.144 | 0.293 | 0.145 | 0.165 | 0.139 | 0.168 | **0.124** | 0.245 | 0.198 | 0.182 | 0.135 | 0.142 | 0.137 | 0.160 | 0.209 | 0.209 |
| M3 (Monthly) | 0.089 | **0.089** | **0.089** | 0.096 | 0.097 | 0.100 | 0.099 | 0.106 | 0.108 | 0.155 | 0.089 | 0.113 | 0.099 | 0.100 | 0.096 | 0.121 | 0.097 | 0.101 | 0.094 | 0.093 | 0.095 | 0.102 | 0.149 | 0.158 |
| M3 (Quarterly) | **0.068** | 0.075 | 0.075 | 0.074 | 0.076 | 0.079 | 0.081 | 0.078 | 0.084 | 0.134 | 0.073 | 0.074 | 0.073 | 0.072 | 0.071 | 0.086 | 0.076 | 0.080 | 0.072 | 0.069 | 0.070 | 0.079 | 0.101 | 0.103 |
| M3 (Yearly) | **0.121** | 0.144 | 0.144 | 0.151 | 0.153 | 0.155 | 0.159 | 0.148 | 0.148 | 0.192 | 0.145 | 0.133 | 0.122 | 0.130 | 0.130 | 0.143 | 0.182 | 0.181 | 0.144 | 0.127 | 0.128 | 0.162 | 0.167 | 0.167 |
| M4 (Quarterly) | 0.077 | **0.062** | 0.062 | 0.082 | 0.083 | 0.084 | 0.086 | 0.087 | - | 0.132 | 0.062 | 0.074 | 0.080 | 0.079 | 0.080 | 0.085 | 0.073 | 0.073 | 0.079 | 0.080 | 0.079 | 0.082 | 0.119 | 0.110 |
| M4 (Yearly) | 0.116 | 0.091 | 0.091 | 0.134 | 0.137 | 0.136 | 0.140 | 0.148 | - | 0.178 | **0.091** | 0.106 | 0.109 | 0.110 | 0.115 | - | 0.114 | 0.118 | 0.115 | 0.130 | 0.161 | 0.161 | | |
| M5 | 0.559 | 0.558 | 0.557 | 0.587 | 0.586 | 0.590 | 0.595 | 0.598 | - | 0.635 | **0.550** | 0.597 | 0.657 | 0.594 | 0.560 | 0.687 | 0.563 | 0.560 | 0.653 | 0.628 | 0.636 | 0.624 | | |
| NN5 (Daily) | 0.152 | 0.158 | 0.155 | 0.156 | 0.161 | 0.169 | 0.173 | 0.162 | 0.242 | 0.261 | 0.152 | 0.149 | 0.155 | 0.154 | **0.145** | 0.159 | 0.149 | 0.147 | 0.293 | 0.264 | 0.294 | 0.312 | 0.425 | 0.425 |
| NN5 (Weekly) | 0.084 | **0.079** | **0.079** | 0.091 | 0.091 | 0.090 | 0.091 | 0.094 | 0.092 | 0.111 | 0.089 | 0.081 | 0.087 | 0.089 | 0.081 | 0.087 | 0.098 | 0.114 | 0.092 | 0.088 | 0.090 | 0.090 | 0.123 | 0.123 |
| Tourism (Monthly) | 0.079 | 0.085 | 0.085 | 0.100 | 0.100 | 0.103 | 0.113 | 0.095 | 0.125 | 0.213 | **0.078** | 0.092 | 0.092 | 0.104 | 0.096 | 0.101 | 0.092 | 0.084 | 0.083 | 0.090 | 0.091 | 0.093 | 0.104 | 0.297 |
| Tourism (Quarterly) | 0.076 | 0.070 | 0.070 | **0.061** | 0.069 | 0.069 | 0.074 | 0.068 | 0.071 | 0.202 | 0.075 | 0.074 | 0.072 | 0.082 | 0.074 | 0.080 | 0.077 | 0.063 | 0.075 | 0.070 | **0.061** | 0.098 | 0.119 | 0.166 |
| Tourism (Yearly) | 0.141 | 0.163 | 0.163 | 0.183 | 0.207 | 0.200 | 0.218 | 0.194 | 0.163 | 0.238 | 0.178 | 0.163 | 0.127 | 0.179 | **0.102** | 0.165 | 0.179 | 0.154 | 0.162 | 0.159 | 0.176 | 0.156 | 0.209 | 0.209 |
| Traffic | 0.243 | **0.164** | 0.213 | 0.256 | 0.264 | 0.263 | 0.264 | 0.254 | 0.287 | 0.256 | 0.170 | 0.246 | 0.233 | 0.234 | 0.264 | 0.290 | 0.263 | 0.270 | - | 0.557 | 0.905 | - | 0.362 | 0.643 |
| Weather | 0.136 | 0.141 | 0.133 | 0.139 | 0.140 | 0.143 | 0.150 | 0.144 | - | 0.164 | **0.127** | 0.143 | 0.147 | 0.152 | 0.151 | 0.174 | 0.143 | 0.144 | 0.174 | 0.214 | 0.217 | 0.185 | 0.217 | 0.217 |
| Geometric Mean (ZS) | **0.585** | 0.699 | 0.680 | 0.634 | 0.649 | 0.655 | 0.667 | 0.675 | 0.784 | 1.134 | 0.624 | 0.689 | 0.745 | 0.876 | 0.627 | 0.762 | 0.659 | 0.671 | 0.720 | 0.821 | 0.757 | 0.751 | 1.000 | 1.157 |
| Geometric Mean (All) | **0.596** | 0.673 | 0.662 | 0.645 | 0.660 | 0.666 | 0.679 | 0.686 | 0.806 | 1.095 | 0.626 | 0.683 | 0.733 | 0.842 | 0.637 | 0.757 | 0.670 | 0.681 | 0.729 | 0.836 | 0.794 | 0.762 | 1.000 | 1.153 |

*Table 8.* MASE and Timing of TimesFM-ICF and TimesFM-FT on the OOD Benchmark.

| Model | Total Fine-Tune Time | Total Inference Time | MASE |
|---|---|---|---|
| TimesFM-ICF | Not needed | 25 min | 0.777 |
| TimesFM-FT (FULL) | 84 min | 28 sec | 0.789 |
| | 167 min | | 0.794 |
| | 250 min | | 0.779 |
| | 334 min | | 0.776 |
| | 418 min | | 0.776 |
| TimesFM-FT (LP) | 29 min | 28 sec | 0.807 |
| | 57 min | | 0.805 |
| | 85 min | | 0.802 |
| | 113 min | | 0.802 |
| | 141 min | | 0.799 |

in-series examples. We measured the validation MASE error of TimesFM-ICF with the number of in-series examples varying from 0-20, and the total number of in-context examples (including randomly selected examples) varying from 1-50. The resulting heatmap is show in Figure 11. The configuration with smallest validation MASE was 11 in-series examples and 34 total examples. The geometric mean MASE ratio (averaged over 5 runs with different random examples selected) was $0.780 \pm .003$ (so within a standard error of the MASE value we report in Figure 5).

While we leave to future work a more detailed investigation of how best to chose relevant examples to add to the context, the results in this paper show that even simple approaches like random selection and selecting examples from the immediate history are sufficient to obtain accuracy gains with in-context fine-tuning.

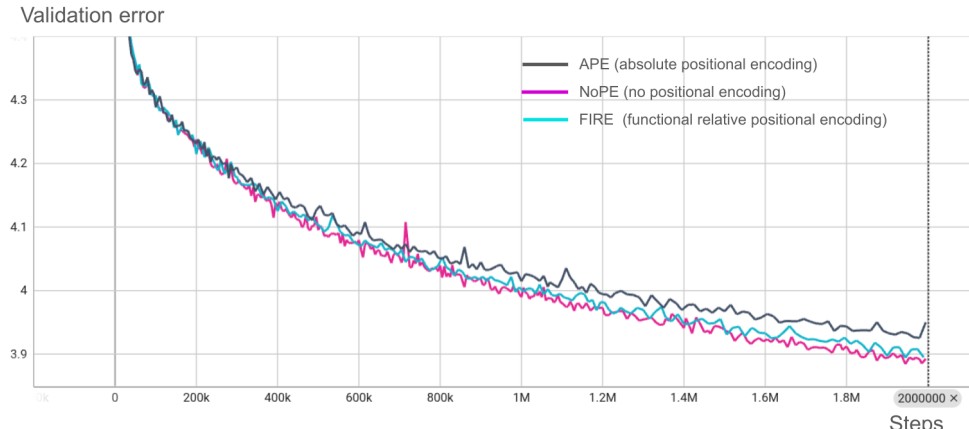

*Figure 9.* Validation errors during training time suggest that (1) NoPE works better than APE, and (2) NoPE performs on par with other positional encodings that generalize length.

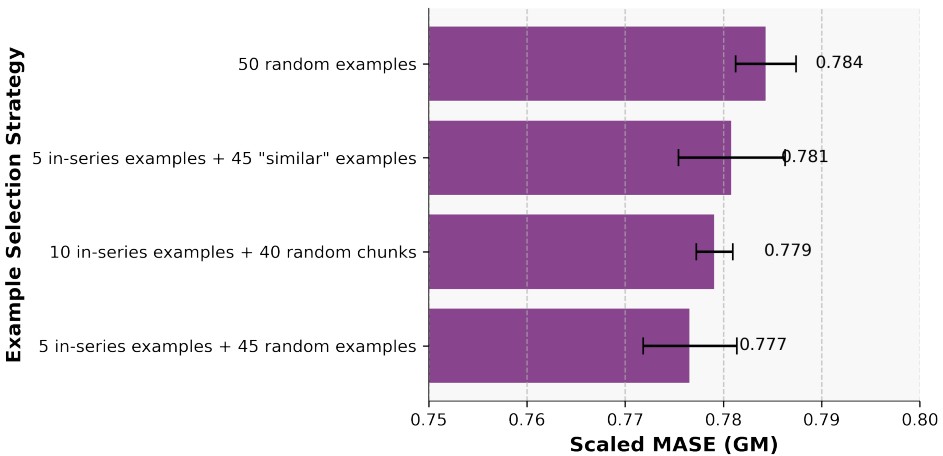

*Figure 10.* Scaled MASE (GM) for various in-context example selection strategies for the OOD benchmark: 1) 50 random examples, 2) 45 Random examples and 5 examples from the immediate past history 3) 45 examples chosen at random from similar time-series (according to DTW distance) and 5 examples from the immediate past history 4) 40 Random examples and 10 examples from the immediate past history. The error bars are one standard deviation of the evaluations averaged over 10 random seeds.

*Table 9.* MAE of TimesFM-ICF against other supervised and zero-shot methods on ETT Rolling Window

| | | Few-shot | | Zero-shot | | | | Task-specific | | | | | | | |
|---|---|---|---|---|---|---|---|---|---|---|---|---|---|---|---|
| Dataset | Horizon | TimesFM-ICF | Moirai-MV (Large) | TimesFM (Base) | Moirai (Small) | Moirai (Base) | Moirai (Large) | TimesFM-FT | iTransformer | TimesNet | PatchTST | Crossformer | DLinear | SCINet | FEDformer |
| ETTh1 | 96 | **0.374** | 0.403 | 0.386 | 0.402 | 0.402 | 0.398 | 0.376 | 0.405 | 0.402 | 0.419 | 0.448 | 0.400 | 0.599 | 0.419 |
| | 192 | **0.400** | 0.432 | 0.409 | 0.419 | 0.429 | 0.434 | 0.401 | 0.436 | 0.429 | 0.445 | 0.474 | 0.432 | 0.631 | 0.448 |
| | 336 | **0.414** | 0.452 | 0.424 | 0.429 | 0.450 | 0.474 | 0.416 | 0.458 | 0.469 | 0.466 | 0.546 | 0.459 | 0.659 | 0.465 |
| | 720 | **0.430** | 0.701 | 0.448 | 0.444 | 0.473 | 0.568 | 0.436 | 0.491 | 0.500 | 0.488 | 0.621 | 0.516 | 0.699 | 0.507 |
| | avg | **0.405** | 0.497 | 0.417 | 0.424 | 0.438 | 0.469 | 0.407 | 0.447 | 0.450 | 0.454 | 0.522 | 0.452 | 0.647 | 0.460 |
| ETTh2 | 96 | 0.327 | 0.341 | 0.337 | 0.334 | 0.327 | 0.325 | **0.325** | 0.349 | 0.374 | 0.348 | 0.584 | 0.387 | 0.621 | 0.397 |
| | 192 | 0.371 | 0.389 | 0.384 | 0.373 | 0.374 | **0.367** | 0.372 | 0.400 | 0.414 | 0.400 | 0.656 | 0.476 | 0.689 | 0.439 |
| | 336 | **0.393** | 0.417 | 0.417 | **0.393** | 0.401 | **0.393** | 0.403 | 0.432 | 0.541 | 0.433 | 0.731 | 0.541 | 0.744 | 0.487 |
| | 720 | 0.422 | 0.507 | 0.446 | **0.416** | 0.426 | 0.421 | 0.424 | 0.445 | 0.657 | 0.446 | 0.763 | 0.657 | 0.838 | 0.474 |
| | avg | 0.378 | 0.414 | 0.396 | 0.379 | 0.382 | **0.377** | 0.381 | 0.407 | 0.497 | 0.407 | 0.683 | 0.515 | 0.723 | 0.449 |
| ETTm1 | 96 | 0.331 | 0.536 | 0.342 | 0.383 | 0.360 | 0.363 | **0.327** | 0.368 | 0.375 | 0.367 | 0.426 | 0.372 | 0.438 | 0.419 |
| | 192 | 0.364 | 0.588 | 0.376 | 0.402 | 0.379 | 0.380 | **0.358** | 0.391 | 0.387 | 0.385 | 0.451 | 0.389 | 0.450 | 0.441 |
| | 336 | 0.387 | 0.625 | 0.402 | 0.416 | 0.394 | 0.395 | **0.381** | 0.420 | 0.411 | 0.410 | 0.515 | 0.413 | 0.485 | 0.459 |
| | 720 | 0.430 | 0.688 | 0.444 | 0.437 | 0.419 | **0.417** | 0.419 | 0.459 | 0.450 | 0.439 | 0.589 | 0.453 | 0.550 | 0.490 |
| | avg | 0.378 | 0.609 | 0.391 | 0.410 | 0.388 | 0.389 | **0.371** | 0.410 | 0.406 | 0.400 | 0.495 | 0.407 | 0.481 | 0.452 |
| ETTm2 | 96 | **0.239** | 0.355 | 0.260 | 0.282 | 0.269 | 0.260 | 0.242 | 0.264 | 0.267 | 0.259 | 0.366 | 0.292 | 0.377 | 0.287 |
| | 192 | **0.283** | 0.418 | 0.306 | 0.318 | 0.303 | 0.300 | 0.284 | 0.309 | 0.309 | 0.302 | 0.492 | 0.362 | 0.445 | 0.328 |
| | 336 | 0.322 | 0.462 | 0.345 | 0.355 | 0.333 | 0.334 | **0.321** | 0.348 | 0.351 | 0.343 | 0.542 | 0.427 | 0.591 | 0.366 |
| | 720 | 0.385 | 0.499 | 0.404 | 0.410 | **0.377** | 0.386 | 0.379 | 0.407 | 0.403 | 0.400 | 1.042 | 0.522 | 0.735 | 0.415 |
| | avg | 0.307 | 0.434 | 0.329 | 0.341 | 0.321 | 0.320 | **0.306** | 0.332 | 0.332 | 0.326 | 0.610 | 0.401 | 0.537 | 0.349 |

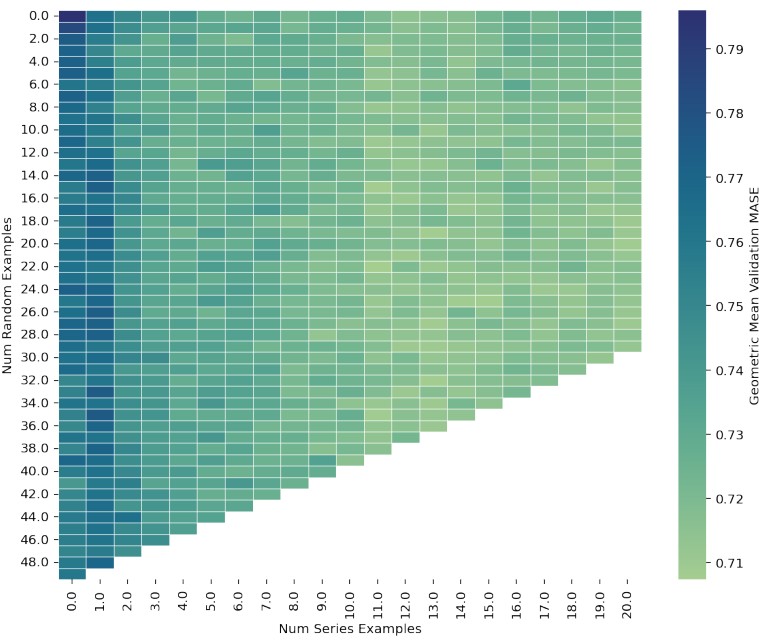

*Figure 11.* Heatmap of in-context example configurations. The configuration with smallest validation loss has 11 in-series examples and 22 randomly-selected examples.

