# OpenReview forum: "In-Context Fine-Tuning for Time-Series Foundation Models"
_ICML.cc/2025/Conference — ICML 2025 poster_

### Official Review · Reviewer_pyrU · 2025-02-19

**Overall Recommendation:** 4

**Summary:**

The paper extends a foundational forecasting models so that it can be conditioned on addition time-series information. Using in-context learning, the values of other time-series and also the value of the time-series to be predicted is added to the model input. Then the model is trained with the initial objective with a modified architecture to handle this additional information. Experiments are conducted on real-world datasets where the method is shown to reach a similar accuracy as fine-tuned models. Compared to those the trade-off is different since no time is spent on the fine-tuning but a larger time is spent at inference.

## edit after rebuttal

I raised my score as the authors addressed my main concerns. Using a validation procedure is I think much cleaner and the heatmap is an interesting addition to understand the benefit of condititioning on more in/out series examples.

**Claims And Evidence:**

Yes, the method performance is validated on out-of-sample datasets. The accuracy is shown to match fine-tuned foundational models.

**Essential References Not Discussed:**

Your work reminded me of Tactis (https://proceedings.mlr.press/v162/drouin22a/drouin22a.pdf), in particular feeding all the time-series dimensions as token to a transformer, it may be worth including it in your related work section.

**Experimental Designs Or Analyses:**

Yes, the author mostly reused benchmarks that have been peer-reviewed and are standard. They also clearly indicates which method performance are in-sample (eg trained on the datasets) and which are not.

**Methods And Evaluation Criteria:**

Yes, a large collection of real-world datasets is considered, standard metrics are reported, and several are used to account for the randomness of the method.

**Other Comments Or Suggestions:**

* l326 Exponential Smoothing (ETS) (missing space)
* toursim

**Other Strengths And Weaknesses:**

Strength:
- impactful application
- sound method that could be applied to other foundational models

Weakness:
- some ablation may be missing
- lack of analysis in the variance and behavior on the conditioning of additional data
- the method makes a trade-off between fine-tuning time and inference time, therefore it is mostly applicable when fine-tuning time is problematic which reduces the scope


-- edit post rebuttal
The authors performed ablations and now use a clear validation protocol.

**Questions For Authors:**

**Analysis and model selection on the Conditioning of extra data.** My biggest complaint on the paper is that there are very few experiments on the conditionning on random data, except for Fig 6. (and Fig 9.) in the appendix and that the process done to select the hyperparameters is not ideal.

The paper is making key design decisions such as selecting some number of in-series and out-series example which appears to be based on the OOD dataset but since you are performing model selection, you should rather use another set of validation datasets to make this selection as you would be overfitting your model selection otherwise.

In addition, I found that the analyses on this aspect lacking, you are reporting the performance on only 4 setup (except in Fig 6 where the number of in-series is varied). Given that the cost of evaluating performance should be reasonable, it would be worth to explore a grid of options for the number of in-series and out-series example and plot a heatmap of the performance. For this, I believe it is a must to consider a set of datasets that are not used for the final selection (or at least to restrict the collections and be explicit about this) to not risk overfitting.

**No variance reported.** Given that your results depend on a random sampling, analysing the noise coming from this decision would be important, it could be done with the analysis I discussed above by reporting confidence interval on mean performance estimates. In addition, reporting the confidence interval on the aggregated results on Fig 5 and others across datasets would be good to convey the uncertainty on the scores.

Those are my biggest concern on the paper, if those points are adressed, I would be keen to revise my score. The other points bellow are not as crucial.

**Additional points**.

- instead of a separator, is there a reason why you did not consider a simple positional encoding? (eg a categorical embedding mapping the time-series index concatenated to your input, of-course using a fix index for the input time-series)
- why are you making a causal mask in your architecture for the conditioning out-series? I would have imagined a mask that allows only the current in-series to look at other out-series but not to let an out-series looks at previous ones as it is less relevant for a out-series to look at the previous ones (also reinforce the random choice of the time-series order when sampling)
- Fig 9: how are you computed the confidence interval? have you considered ensembling the predictions obtained when sampling different history?

**Relation To Broader Scientific Literature:**

Related work is well cited.

**Theoretical Claims:**

NA

---

> ### Author Rebuttal · Authors · 2025-04-01
>
> We thank the reviewer for their helpful comments and suggestions. We address the main points below. Please let us know if you have any further comments or concerns. If our response sufficiently addresses your concerns, we hope that you will consider updating your score accordingly.
>
> > Given that the cost of evaluating performance should be reasonable, it would be worth to explore a grid of options for the number of in-series and out-series example and plot a heatmap of the performance. For this, I believe it is a must to consider a set of datasets that are not used for the final selection (or at least to restrict the collections and be explicit about this) to not risk overfitting.
>
> Thank you for this feedback. Based on your suggestion, we constructed a validation dataset from the training portion of a subset of the Monash datasets (specifically: weather, traffic, australian electricity, ercot, ETTm, and ETTh). We chose these datasets because they contained many training examples long enough to construct up to 20 in-series examples. We measured the validation MASE error of TimesFM-ICF with the number of in-series examples varying from 0-20, and the total number of in-context examples (including randomly selected examples) varying from 1-50. The resulting heatmap is attached in Figure 1 in this link: https://anonymous.4open.science/r/icml25-DB6C/icml25.pdf. The configuration with smallest validation MASE was 11 in-series examples and 34 total examples. The geometric mean MASE ratio (averaged over 5 runs with different random examples selected) was 0.780+/.003 (so within a standard error of the MASE value we report in Figure 5).
>
> > Given that your results depend on a random sampling, analysing the noise coming from this decision would be important, it could be done with the analysis I discussed above by reporting confidence interval on mean performance estimates. In addition, reporting the confidence interval on the aggregated results on Fig 5 and others across datasets would be good to convey the uncertainty on the scores.
>
> Please see updated OOD tables in https://anonymous.4open.science/r/icml25-DB6C/icml25.pdf which now include confidence intervals for TimesFM-ICF. Each entry in the tables is an average over 5 runs of random in-context example selection (using our original in-series configuration). Note that our Figures already include confidence intervals.
>
> > instead of a separator, is there a reason why you did not consider a simple positional encoding? (eg a categorical embedding mapping the time-series index concatenated to your input, of-course using a fix index for the input time-series)
>
> Using a time series level positional encoding is definitely a valid option. One advantage of our proposal is that the pretrained model weights could potentially generalize to more than 50 in-context examples thanks to our current choice of not using any form of positional encoding. This is an open question we will empirically explore in the future.
>
> > why are you making a causal mask in your architecture for the conditioning out-series? I would have imagined a mask that allows only the current in-series to look at other out-series but not to let an out-series looks at previous ones as it is less relevant for a out-series to look at the previous ones (also reinforce the random choice of the time-series order when sampling)
>
> As we train the model in decoder-only manner, during training time there is no notion of a single “in-series”: for the forward pass on a training context (see Section 5.1. Context Generation), every series (example) is its own “in-series” and all its preceding series are its “out-series”, therefore we have to apply a full context level causal attention so that decoder-only works. At inference time we hence choose to keep the attention between out-series, which is for consistency with model training. This is identical to how language models commonly handle in-context examples (few-shot prompts).
>
> It is a good point that our design breaks the symmetry among out-series. Empirically we tried to minimize its effect by training with randomizing the order of in-context examples when there is no causal leakage (see Section 5.1. Context Generation, grouping, “Dataset level:”). It’s also an interesting question that what happens if we remove this attention between out-series at inference time - we will do it as a future study.
>
>
> > Fig 9: how are you computed the confidence interval? have you considered ensembling the predictions obtained when sampling different history?
>
> In Fig 9, the uncertainty of the reported metric comes from the random selection of in-context examples, and the confidence interval is computed based on 10 runs of the same setup with different random seeds.
>
> It’s definitely possible to ensemble the predictions based on repeated sampling of different in-context examples. The practitioner can make this choice at the cost of increased latency.

---

> > ### Comment · Reviewer_pyrU · 2025-04-02
> >
> > Thank you for your answer and additional experiments, I will raise my score as it addressed my main concerns.
> > Using a validation procedure is I think much cleaner and the heatmap is an interesting addition to understand the benefit of condititioning on more in/out series examples.
> >
> > I have two small remaining suggestions:
> >
> > > In Fig 9, the uncertainty of the reported metric comes from the random selection of in-context examples, and the confidence interval is computed based on 10 runs of the same setup with different random seeds.
> >
> > It would be great to put this in the paper as it was not mentioned as far as I could see (I may have missed it).

---

> > > ### Author Response · Authors · 2025-04-07
> > >
> > > Thank you for your reply and additional suggestions. We will incorporate this into the final version.

---

### Official Review · Reviewer_U55B · 2025-02-24

**Overall Recommendation:** 5

**Summary:**

This paper proposes a novel in-context finetuning strategy for a specific Time Series Foundation Model (TSFM). By continual pretraining a TSFM (TimesFM in the paper) with in-context examples, the updated model is able to be prompted with related past time series examples at inference time, enhancing forecasting accuracy without requiring additional training pipelines. The main contribution is to adapt the idea of "in-context learning" in NLP domain to TSFM. Experimental results validate the effectiveness of the proposed approach across multiple forecasting tasks.

**Claims And Evidence:**

Most of the claims are reasonable. However, some may not be clearly supported—please refer to *Experimental Designs or Analyses* and *Other Strengths and Weaknesses* sections for details.

**Essential References Not Discussed:**

Related works are properly cited and discussed in the paper.

**Experimental Designs Or Analyses:**

Most experimental designs and analyses make sense, except the following ones.

* *Section 6.3*: The design of using Moirai for in-context examples appears problematic. The key issue with this experiment is that **past related examples cannot be used as multivariate features in Moirai**. In Moirai’s framework, multivariate features correspond to **different variables within the same example**, sharing the same temporal range and aligned time ID. However, treating past related examples as additional multivariate features introduces a mismatch, as they belong to entirely different temporal periods. This misalignment causes Moirai’s time ID to convey misleading information, leading to the extraction of meaningless temporal relationships via attention. Consequently, this likely explains why Moirai-MV’s performance degrades when additional multivariate features are introduced in your experiments. In summary, the comparison with Moirai-MV in this manner is confusing and potentially misleading.

* *Section 6.4.2*:  TimesFM (base) has a maximum history length of 512 and relies on positional encoding. Thus, I don't think directly inferencing with a longer history ($L=2048$) would significantly improve performance without additional training. Notably, TimesFM-ICL undergoes a continual pretraining phase. A fair comparison would involve continually pretraining TimesFM (base) with a longer history length ($L=2048$) before evaluating its performance against TimesFM-ICL. In my view, the current evaluation may potentially undervalue the impact of a longer history in TimesFM.

**Methods And Evaluation Criteria:**

The proposed methods and evaluation criteria make sense for the problem.

**Other Comments Or Suggestions:**

* Suggestions: The paragraph **In-Context Example Selection** in Section 6.2 should be moved to an earlier section if it is a fundamental component of the framework. Placing it within the experimental results section appears confusing and incoherent:  it is unclear whether this in-context example selection strategy is applied during both continual pretraining and testing or only during inference in this specific experiment.
* Typos: In the title of Figure 1, it should be _left/right_ instead of _top/bottom_.

**Other Strengths And Weaknesses:**

Strengths:
1. The exploration of in-context learning for TSFM is pioneering and important.
2. The paper is generally well-written and clearly structured.
3. Extensive experiments are conducted, providing a comprehensive analysis of the proposed method.


Weakness:
1. In-context fine-tuning appears to be highly inefficient in terms of inference time. According to Table 7, the total inference time of TimesFM-ICF is 25 minutes—almost 50 times slower than TimesFM without ICF. The authors argue that fine-tuning is inefficient by taking training time into account; however, there are two key issues with this claim:
    * In-context fine-tuning also requires an additional continual pretraining phase, yet the authors do not account for this computational cost.
    * When compared to TimesFM (zero-shot), whose total inference time should be similar to TimesFM-FT, TimesFM-ICF offers no efficiency advantage at all.

    This raises a critical question: is the forecasting improvement provided by in-context fine-tuning worth such a significant trade-off in inference speed?

**Questions For Authors:**

1. Is the proposed in-context fine-tuning method generalizable to other TSFMs beyond TimesFM?
2. Why is the maximum history length in TimesFM limited to 512? What constraints impose this limit—positional encoding, architectural consideration, or just pretraining configurations?
3. Why can TimesFM-ICF handle only a maximum of 50 examples in its context? Is this limitation due to the continual pretraining process being designed specifically for 50 prompted examples, or are there other underlying constraints?
4. I believe Moirai is trained to accommodate a maximum of 128 variables, not 50 as stated in Line 353. Could you clarify this discrepancy?
5. What is the fundamental difference between in-context fine-tuning and using exogenous variables? Can the ICF method be interpreted as treating past examples as exogenous variables?
6. Why the length of in-context examples can be different? Aren't they in the identical length of $L+h$?

**Relation To Broader Scientific Literature:**

The key contribution of this work relates to in-context learning or few-shot learning in the LLM domain. It aims to enable TSFM to leverage a few prompted examples at inference time, enhancing forecasting performance without additional training. This work represents a pioneering effort in exploring this direction.

**Theoretical Claims:**

Nothing to discuss as no proofs for theoretical claims are included in the paper.

---

> ### Author Rebuttal · Authors · 2025-04-01
>
> We thank the reviewer for their helpful comments and suggestions. We address the main points below. Please let us know if you have any further questions. If our response addresses your concerns, we hope you consider raising your score accordingly.
>
> > Section 6.3: Moirai
>
> Thank you for clarifying this point. Our interpretation from the Moirai paper in Section 3.2.2 Task Distribution was that the training datasets were augmented by “constructing multivariate time series from sub-datasets with univariate time series, by randomly concatenating them”. Based on this, it seemed that Moirai could support past related examples as multivariate features. We do not want to misrepresent the capabilities of the Moirai model in our paper, so we will contact the Moirai authors for clarifications, and correct this as the reviewer suggests.
>
>
> > Section 6.4.2: TimesFM (base)
>
> We apologize for the lack of clarity - we indeed pretrained TimesFM-LH with a longer history length of 2048 (in a manner similar to the latest version of the TimesFM repo (https://huggingface.co/google/timesfm-2.0-500m-pytorch). We didn't just use TimesFM (base) directly with a 2K context length at inference time to get the TimesFM-LH results. We will make this more clear in Section 6.4.2
>
> > ICF appears to be highly inefficient in terms of inference time [...]
>
> Since the total context length for TimesFM-ICF is 50 times larger than TimesFM, the 50 times inference speed slowdown is not unreasonable. However, TimesFM-ICF is still a zero-shot model, the addition continual pretraining phase for training TimesFM-ICF is a one-time cost that is independent of the target dataset – there is no cost to be paid to adapt the model for a new domain or dataset (unlike fine-tuning costs, which has to be paid to adapt TimesFM for every new dataset). More importantly, for practitioners, the ability to use the ICF model out of the box, and avoid having to build a fine-tuning pipeline to customize the model for their use-case is a very significant cost and resource advantage that we think can more than offset the 50x inference speed disadvantage (note that even after this 50x inference slowdown, TimesFM ICF can still perform inference for a single example in 43 ms on average - totalling 25 minutes on approximately 140k testing time series on TPUv5e with 4 cores -  which is often sufficient for most practical forecasting use-cases)
> > Suggestions [Sec 6.2]
>
> Thank you for the suggestions. We will update the final version of our paper accordingly.
>
> > other TSFMs beyond TimesFM?
>
> We believe that our methodology is generalizable to other TSFMs. This is an interesting direction for future research.
>
> > maximum history length limited to 512?
>
> The maximum history length constraint in TimesFM comes because of the pretraining configuration - the model has been pretrained on examples up to the maximum history length and might not generalize well to lengths beyond what it has been pretrained on. Increasing the maximum history length during pretraining will result in longer training times and compute resources
>
> > maximum of 50 examples in context?
>
> The choice of 50 examples for TimesFM-ICF comes from its continual pretraining setup where the model is trained with up to 25K (512*50) context window. Increasing the context window in continual pretraining would have caused longer training time and compute resources, which we would not have been able to afford. At inference time the model may generalize to beyond 50 examples - there is no mechanism within the model casting a limit of 50 indeed. For simplicity and conciseness of the paper, we also use 50 examples in our empirical study for consistency with the continual pretraining setup. Verifying the generalization is a separate, empirically heavy task that we plan to study in the future.
>
> > I believe Moirai is trained to accommodate a maximum of 128 variables, not 50
>
> Sorry for the wording issue. We did not intend to imply that it accommodates at most 50, just that it accommodates 50. We will clarify our wording in the final version.
>
> > What is the fundamental difference between in-context fine-tuning and using exogenous variables?
>
> We view in-context fine-tuning and exogenous variables as two separate concepts: The former supplements the main forecasting history with time series generated from the same or a similar distribution, in which sense it is in-context “fine-tuning” to fit this distribution. On the other hand, exogenous variables can be of any distribution, and their usefulness mainly comes from their correlation with the main forecast task. In this regard, in-context fine-tuning is a more strict practice.
>
> > Why the length of in-context examples can be different?
>
> At both training and inference time we apply paddings to short in-context examples to bring them to length L+h while at the same time apply proper attention masks. Please see Section 5.1., Context Generation.

---

> > ### Comment · Reviewer_U55B · 2025-04-02
> >
> > Thank you for the detailed reply. All points and concerns have been properly discussed and clearly explained by the authors. Regarding the Moirai-MV issue, please provide an update here once you hear back from the Moirai authors. Once again, I believe the quality of this work is solid, and I will update my scores accordingly. Please remember to make the corresponding revisions in your final version.

---

> > > ### Author Response · Authors · 2025-04-07
> > >
> > > Thank you for your reply.
> > >
> > > We wanted to provide an update after contacting the Moirai authors. They clarified that none of their training data contains multivariate time-series with covariates from different time windows. Based on this, and on your response, we agree that our current comparison with Moirai-MV without any caveats is misleading. Of course, this was not our intention, so we will correct this in the final version of our paper.
> > >
> > > We plan to remove the Moirai-MV column from Table 1 in the main body to eliminate this confusion. We will move the Moirai-MV result to the appendix, and emphasize there that the performance degradation is likely due to misalignment of the Time ID. We think it is useful to include this result in the appendix (with this caveat) to demonstrate that naively concatenating related time-series as multivariate features may not work out-of-the-box.
> > >
> > > Please let us know if this seems like a reasonable modification to you.

---

### Official Review · Reviewer_pqjj · 2025-03-09

**Overall Recommendation:** 4

**Summary:**

The paper proposes a "fine-tuning strategy via in-context learning" for pre-trained time series forecasting models. Essentially, the approach is similar to few-shot learning in LLM as multiple time series are added to the context in addition to the forecasted context.
The authors modify an existing architecture and introduce a separation token to utilize multiple series in one context.
Experiments suggest that the approach is effective and improves the forecasting performance of the pre-trained model.

## update after rebtuall
we thank the reviewer for clarification. i think the paper is valuable and i keep my score at accept

**Claims And Evidence:**

The main claim that the approach can improve the performance of pre-trained time series models is well supported by the empirical results.

**Essential References Not Discussed:**

I am not aware of any essential references that are missing.

**Experimental Designs Or Analyses:**

The experiments in section 6 seem appropriate and sound.
The selected benchmark data is especially a good choice as it is more comprehensive than a lot of other work.
The ablation experiments are reasonable and give important insights to justify the approach (comparison to long context model) as well as the specific in-context sample selection procedure.
Regarding the "in-context sample selection procedure", I would suggest to further add the performance variation over the runs over the individual datasets (similar to Figure 9 for the overall results).

**Methods And Evaluation Criteria:**

The method is well motivated by the success of in-context learning in LLM.
A comprehensive benchmark and standard metrics are utilized for evaluation and, therefore, make sense.

**Other Comments Or Suggestions:**

On Page 2 / line 259 the authors refer to A.2. although the authors probably want to refer to A.5

**Other Strengths And Weaknesses:**

Strength:
- The approach is not dependent on the specific architecture but could be applied to other pre-trained time series architectures.
- Approach reaches fine-tuning performance
- Evaluation on a comprehensive benchmark

Weakness:
- Only the subset of zero-shot benchmark is utilized. While this is a good idea to preserve the "zero-shot" setting, it would be advisable to additionally report results for the full benchmark and report which datasets are non zero-shot as this would make comparison to other papers/models easier.
- As the evaluation includes randomness of the in-context samples, the authors should provide the variation over the individual datasets of the benchmark similar to Figure 9 for the "overall result".
- Context length increases linearly with an increasing amount of context samples, therefore, computation complexity for the transformer increases quadratically

**Questions For Authors:**

- Why did you decide to call the benchmark OOD benchmark? Typically, OOD is more often referred to situations which one has to detect (OOD detection) as the models do not perform reliable on it. In my understanding, the idea of pre-trained models is that the models should generalize over time series data, including the evaluated zero-shot data. Hence, in most related literature, this evaluation is more often referred to as "zero-shot" benchmark.

**Relation To Broader Scientific Literature:**

The approach is embedded in the literature of pre trained time series models, more specifically in pretraine time series forecastin models.
The contribution can improve the performance of such models in general, as it does not necessarily depend on a certain architecture. Further, it might extend the potential use case scenarios for these models
Further, it is related to in-context learning in general, which became especially popular for LLM.

**Theoretical Claims:**

There are no theoretical claims or proofs

---

> ### Author Rebuttal · Authors · 2025-04-01
>
> We thank the reviewer for their helpful comments and suggestions. We address the main points below. Please let us know if you have any further comments or concerns.
>
> > Only the subset of zero-shot benchmark is utilized. While this is a good idea to preserve the "zero-shot" setting, it would be advisable to additionally report results for the full benchmark and report which datasets are non zero-shot as this would make comparison to other papers/models easier.
>
> Thank you for the suggestion. We have added evaluations on the missing four datasets (m4 quarterly, m4 yearly, weather, and traffic) in Table 1 (MASE) and Table 2 (WQL) here: https://anonymous.4open.science/r/icml25-DB6C/icml25.pdf. In these tables, we additionally report the geometric means of the scores over the 23 (zero-shot) datasets originally reported in our paper (row “Geometric Mean (ZS)”) and over all 27 datasets (“Geometric Mean (All)”).
>
> > As the evaluation includes randomness of the in-context samples, the authors should provide the variation over the individual datasets of the benchmark similar to Figure 9 for the "overall result".
>
> We have also added confidence intervals to Tables 1 and 2 in the link above (https://anonymous.4open.science/r/icml25-DB6C/icml25.pdf). Note that all of our figures already include error bars. It was not clear enough in Figure 5 and we will fix it.
>
> > Context length increases linearly with an increasing amount of context samples, therefore, computation complexity for the transformer increases quadratically
>
> We agree with the reviewer’s comment here. This quadratical attention computational complexity does not directly translate to a quadratical inference time likely due to (1) the latency of the feedforward layers and (2) optimization of transformer implementation on modern accelerators.
>
> > Why did you decide to call the benchmark OOD benchmark? Typically, OOD is more often referred to situations which one has to detect (OOD detection) as the models do not perform reliable on it. In my understanding, the idea of pre-trained models is that the models should generalize over time series data, including the evaluated zero-shot data. Hence, in most related literature, this evaluation is more often referred to as "zero-shot" benchmark.
>
> Thanks for pointing this out. We have abused the notion of OOD as out of the pretraining data distribution. We did not call it “zero-shot benchmark” because such a name was mentioned in [1] and we wanted to be clear that our benchmark is its subset that’s zero-shot for TimesFM-ICF.  We will clarify and update the name to zero-shot benchmark.
>
> [1] Ansari, Abdul Fatir, et al. "Chronos: Learning the Language of Time Series." Transactions on Machine Learning Research.

---

### Official Review · Reviewer_BzCE · 2025-03-13

**Overall Recommendation:** 3

**Summary:**

The authors propose a framework to obtain pretrained models for time series forecasting that are capable of doing in-context learning. The authors approach is verified on top of TimesFM (a decoder only pretrained model for time series forecasting) accompanied with extensive evaluations. The authors show that the proposed approach not only outperforms data-specific models and other pretrained models in zero-shot settings, but that it even outperforms fine-tuning variants of the proposed model.

**Claims And Evidence:**

The main claim of the paper is that the proposed approach is a direct extension of NLP pretrained models to the field of time series forecasting. Whereas in NLP in-context-learning has shown to be a relevant property, it remains as an open question in time series if this can be extended. The authors claim that the proposed approach provides a positive answer to this question.

Perhaps one of the most relevant points of the paper is that the extension is relatively easy. It is based on adding extra tokens that represent a split between consecutive in-context-samples, so that the model is able to identify that different entities are provided in the context, which can be used to improve the quality of the generated forecast.

Further, the authors provide extensive empirical evidence on the relevance and methodology to identify in-context-samples. They show that indeed the more samples are provided, the better the performance, albeit at the expense of larger inference time.

The paper claims are well sustained with extensive empirical evidence. I have a couple of questions about the methodology, but I believe this can be addressed somehow by the authors, and this do not hinder the interesting contribution done in the paper.

**Essential References Not Discussed:**

The authors do a fair job in comparing with other state of the art models.

**Experimental Designs Or Analyses:**

The experimental design is methodologically sound. The authors provide interesting studies on the effect of how the in-context samples can be chosen (time-series level and Dataset level), and the amount of samples to be chosen (the larger the better at the expense of larger inference time), and competitive performance against full-fine tuning of the model. In general the authors did emphasis in that no data-leakage happens when doing out-of-domain evaluations.

**Methods And Evaluation Criteria:**

**Evaluation**. The authors provide evaluations in terms of MASE, which assume that the considered models provide only point forecasts. Several of the models here considered are able to provide probabilistic forecasts, and at least in the original TimesFM paper it was claimed that it would be possible to generate probabilistic forecasts as well. Moreover, since the authors main reference for evaluation setup is Ansari et al, which introduces Chronos (a probabilsitic pretrained model), I wonder if the provided model can generate probabilistic forecasts and what the corresponding evaluations in terms of mean weighted quantile loss would be.

**Datasets**. The authors provide extensive evaluations. Although the authors take as reference the experimental set up of Ansari et al, the authors decided to evaluate in 23 of 27 of the datasets. This decision seems to have been made because the excluded datasets where used for pretraining the proposed model. This makes it unclear if the main results of Figure 5 hold if these datasets were included in the zero-shot evaluation.

**Qualitative Evidence**. The paper is missing visualizations of the generated forecasts. It would be great to see if there are any visual clear changes of the generated forecasts when using a larger number of in-context-samples.

**Other Comments Or Suggestions:**

I would suggest the authors to update the bibliography entries. Several papers are already accepted at top conferences, and still the authors cite them with their Arxiv entries. Examples of this are:

- Ansari, A. F., Stella, L., Turkmen, C., Zhang, X., Mercado,
P., Shen, H., Shchur, O., Rangapuram, S. S., Arango, S. P.,
Kapoor, S., et al. Chronos: Learning the language of time
series. arXiv preprint arXiv:2403.07815, 2024.

- Goswami, M., Szafer, K., Choudhry, A., Cai, Y., Li, S., and
Dubrawski, A. Moment: A family of open time-series
foundation models. arXiv preprint arXiv:2402.03885, 2024.

- Gruver, N., Finzi, M., Qiu, S., and Wilson, A. G. Large
language models are zero-shot time series forecasters.
arXiv preprint arXiv:2310.07820, 2023.

- Haviv, A., Ram, O., Press, O., Izsak, P., and Levy, O.
Transformer language models without positional encodings still learn positional information. arXiv preprint
arXiv:2203.16634, 2022.

- Li, S., You, C., Guruganesh, G., Ainslie, J., Ontanon, S.,
Zaheer, M., Sanghai, S., Yang, Y., Kumar, S., and Bhojanapalli, S. Functional interpolation for relative positions improves long context transformers. arXiv preprint
arXiv:2310.04418, 2023.

- Liu, P. J., Saleh, M., Pot, E., Goodrich, B., Sepassi, R., Kaiser, L., and Shazeer, N. Generating
wikipedia by summarizing long sequences. arXiv preprint
arXiv:1801.10198, 2018.

- Zhou, T., Niu, P., Wang, X., Sun, L., and Jin, R. One fits
all: Power general time series analysis by pretrained lm.
arXiv preprint arXiv:2302.11939, 2023.

**Other Strengths And Weaknesses:**

See above.

**Questions For Authors:**

1. why in the proposed paper the base model is trained on even more datasets (LOTSA datasets) than it was in the original TimesFM paper? I wonder if the results would still hold if the same pretraining methodology was followed as in the original TimesFM paper.
2. if the authors are pretraining the base model, why did they decide to exclude certain datasets that could be used for zero-shot evaluations as in Ansari et al? Reading the supplementary material (Section A.5 -- lines 678-681), for some of these cases maybe the context length would have been problematic, but it seems that even more standard datasets like traffic and weather were excluded from the zero-shot evaluation. Whereas I believe that the contribution of the authors is strong enough to not be hindered by these decisions, these issues open questions rather than clarity.
3. In the original TimesFM paper one of the mentioned limitations is that covariates can not be included in the pretrained model. I wonder if the proposed approach in this paper can accomodate for this. Perhaps in-context-samples can be taken from covariates that evolve on time (dynamic features). This would be a clear-cut for practitioners to adopt this model in the industry, as covariates often come as de-facto request.

**Relation To Broader Scientific Literature:**

The paper is well relevant in the field of time series forecasting, and it is comparing with state of the art models.

**Theoretical Claims:**

There are not theoretical claims.

---

> ### Author Rebuttal · Authors · 2025-04-01
>
> We thank the reviewer for their helpful comments and suggestions. We address the main points below. Please let us know if you have any further comments or concerns. If our response sufficiently addresses your concerns, we hope that you will consider raising your score accordingly.
>
>
>
> > I would suggest the authors to update the bibliography entries.
>
>
> Thank you for pointing this out. We will update the bibliography accordingly in the final version of our paper.
>
>
> > why in the proposed paper the base model is trained on even more datasets (LOTSA datasets) than it was in the original TimesFM paper? I wonder if the results would still hold if the same pretraining methodology was followed as in the original TimesFM paper.
>
>
> While our study focuses on the improvement introduced by the in-context fine-tuning continued training over a base model, we intend to start from an up-to-date base model. Therefore we use the same datasets that are specified in the latest version (v2) of the TimesFM huggingface repo (https://huggingface.co/google/timesfm-2.0-500m-pytorch), which does include LOTSA.
>
>
> > In the original TimesFM paper one of the mentioned limitations is that covariates can not be included in the pretrained model. I wonder if the proposed approach in this paper can accomodate for this. Perhaps in-context-samples can be taken from covariates that evolve on time (dynamic features). This would be a clear-cut for practitioners to adopt this model in the industry, as covariates often come as de-facto request.
>
>
> As our model is trained only with in-context examples coming from the same or a similar time-series that shares the same / a similar distribution as the target time series, we expect that providing other covariates as in-context examples likely would not yield good performance out-of-the-box. Adapting our model to allow for additional covariates is an interesting direction for future work.
>
>
> > I wonder if the provided model can generate probabilistic forecasts and what the corresponding evaluations in terms of mean weighted quantile loss would be.
>
>
> Thank you for your suggestion. We have evaluated the wQL from our proposed model along with some of the other baseline methods, and the results show that our model performs significantly better on wQL compared to the baselines. The full results are uploaded separately in Table 2 here: https://anonymous.4open.science/r/icml25-DB6C/icml25.pdf, and we will add these results to the next version of the paper.
>
>
> > Missing visualization.
>
>
> Thanks for pointing out our lack of clarification here. In the current paper, Fig. 7 is a limited visual example of how the ICF model behaves differently from a base model. We’ve created additional visualization on the australian_electricity dataset (which has 5 time series) to demonstrate how the forecast changes with a large number of in-context examples. Refer to Figure 2 in the following link: https://anonymous.4open.science/r/icml25-DB6C/icml25.pdf. In this figure, we plot the predictions of TimesFM-ICF operating in three modes: 0 in-context examples, 20 (random) in-context examples, and 50 in-context examples (5 of which are within-series examples). These three configurations have increasingly better MASE scores on this dataset (with MASE values ~1, ~.9, and ~.8, respectively). The predictions visually appear to improve with the MASE values.
>
> > Report on 27 datasets instead of 23
>
> We have added evaluations on the missing four datasets (m4 quarterly, m4 yearly, weather, and traffic) in Table 1 (MASE) and Table 2 (WQL) here: https://anonymous.4open.science/r/icml25-DB6C/icml25.pdf. In these tables, we additionally report the geometric means of the scores over the 23 (zero-shot) datasets originally reported in our paper (row “Geometric Mean (ZS)”) and over all 27 datasets (“Geometric Mean (All)”). The results of Figure 5 continue to hold when adding the additional 4 datasets (which recall are not zero-shot for our model).

---

### Decision · Program_Chairs · 2025-05-01

**Decision:**

Accept (poster)

**Comment:**

This paper proposes a methodology for in-context fine-tuning of a time-series foundation model, allowing the model to exploit multiple support series at inference without further training. Tested on multiple public datasets, it yields consistent zero‑shot improvements. The authors also demonstrate the performance of the methodology through various ablation studies on context size and selection, plus extra validation provided during the rebuttal stage. Overall, all reviewers agree on the idea’s novelty, simplicity, and generality. Therefore, I recommend acceptance of this paper.